# PREBLE: EFFICIENT DISTRIBUTED PROMPT SCHEDULING FOR LLM SERVING

**Vikranth Srivatsa**[*1]**, Zijian He**[*1]**, Reyna Abhyankar**[1]**, Dongming Li**[1]**, Yiying Zhang**[1,2]
[1]University of California, San Diego, [2]GenseeAI Inc.

## ABSTRACT

Prompts to large language models (LLMs) have evolved beyond simple user questions. For LLMs to solve complex problems, today's practices are to include domain-specific instructions, illustration of tool usages, and/or long context such as textbook chapters in prompts. As such, many parts of prompts are repetitive across requests. Recent works propose to cache and reuse KV state of prompts. However, they are all confined to a single-GPU optimization, while production LLM serving systems are distributed by nature.

This paper proposes Preble, the first distributed LLM serving platform that targets and optimizes for prompt sharing. We designed a distributed scheduling system that co-optimizes KV state reuse and computation load-balancing with a new scheduling algorithm and a hierarchical scheduling mechanism. Our evaluation of Preble with real workloads and request arrival patterns on two open-source LLMs shows that Preble outperforms the SOTA serving systems by $1.5\times$ to $14.5\times$ on average latency and $2\times$ to $10\times$ on p99 latency.

## 1 INTRODUCTION

Recently, new capabilities and use cases of LLMs have created two common features not seen in traditional LLM usages. First, prompts to LLMs are significantly longer than generated sequences. For example, questions about a long document (Li et al., 2023a) or a video clip Xiao et al. (2021) are answered by LLMs with short answers. As another example, detailed instructions and illustrations for LLMs are vital in accomplishing complex tasks like solving advanced math problems Yao et al. (2023b). The latest LLMs like OpenAI o1 have built-in reasoning capabilities; even though the end users' prompts do not need to be long, these models internally add more context like chain-of-thought prompting Wei et al. (2024), lengthening the total context length OpenAI (2024a). As demonstrated by o1 and other model usage, oftentimes, the longer a prompt is, the better quality the model can generate. Long prompts with short generations imply that the prefill phase significantly outweighs the decoding phase. Thus, improving the prefill phase performance is crucial to the overall performance of LLM serving systems.

Second, prompts are partially shared across requests. For example, a long document or video is often queried many times with different questions Li et al. (2023a); different requests using the same tools share tool instructions in tool-augmented LLMs Hao et al. (2023); chain- or tree-structured prompting calls an LLM in steps, with each subsequent step reusing context from previous steps Yao et al. (2023b); Zhang et al. (2023); Yao et al. (2023a). Real-world production systems like Anthropic have also reported the wide existence of long and shared prompts Anthropic (2024).

Recent works Zheng et al. (2023b); Gim et al. (2024) propose to cache computed key-value (KV) state in GPU memory and reuse the cached KV when a new request sharing a prompt prefix arrives. These works aim to improve the serving performance of LLMs with long and shared prompts in a *single model* replica.

---

[*]Equal contribution

However, in-production LLM serving systems typically utilize a distributed set of GPUs to serve user requests. Current distributed LLM serving systems are not prompt-cache-aware; they attempt to distribute LLM computation load equally across GPUs to achieve high cluster-level GPU utilization. Yet, this distribution could result in requests with shared prefixes being sent to different GPUs, causing KV computation at all these GPUs that could otherwise be avoided if prefixes are cached and reused on the same GPU. On the other hand, a naive solution that always sends requests with shared prefixes to the same GPU would result in imbalanced loads and low overall GPU utilization because the GPU that initially serves a request with a popular prefix will accumulate a huge load of new requests all trying to reuse the calculated prefix KV.

To properly design a distributed LLM serving system for long and shared prompts, we first perform a comprehensive study of five typical LLM workloads: LLM with tool calling Guo et al. (2024), LLM as embodied agents in virtual environments Huang et al. (2022), LLM for code generation Nijkamp et al. (2023), embedded video QA Xiao et al. (2021), and long document QA Li et al. (2023a). We find their prompts to be $37\times$ to $2494\times$ longer than generated sequences, and 85% to 97% tokens in a prompt are shared with other prompts. Additionally, a request often shares prompts with multiple other requests at different amounts (*e.g.*, one prefix sharing being the initial system prompt, one sharing being the system prompt plus tool A's demonstration, and one being the system prompt plus tool A and tool B's demonstrations). We also analyze the Azure LLM request trace Patel et al. (2024) and found these end-user traces to have large prompt-to-output ratios as well. Additionally, this trace shows that requests arrive at varying speeds over time and across LLM usages, making designing a distributed LLM serving system challenging.

Based on our findings, we propose a distributed LLM request scheduling algorithm called *E2* (standing for *Exploitation + Exploration*) that co-designs model computation load-balancing and prefix-cache sharing. E2 allows requests to *exploit* (*i.e.*, reuse) computed prompt prefixes on the same GPU but also gives chances for requests with shared prefixes to *explore* other GPUs. E2 chooses exploitation when the amount of recomputation saved is larger than that of new computation, which happens when the number of shared prefix tokens is larger than the remaining non-shared tokens. Otherwise, E2 chooses exploration. For exploitation, we send the request to the GPU that caches the longest-matched prefix.

When E2 decides to explore GPUs, it chooses the GPU with the lightest "load", using a *prompt-aware load* definition we propose. This prompt-aware load includes three parts all calculated as GPU computation time. The first part is a GPU's computation load in a recent time window $H$, which is measured by the total prefill time and decode time incurred by all the requests in $H$. The second part is the cost of evicting existing KVs on the GPU to make memory space to run the new request. The third part is the cost of running the new request on the GPU. When calculating these three parts, we account for prompt sharing that have or would occur on the GPU by treating the reusable computation as zero cost. E2 picks the GPU with the lowest sum of the three parts to explore, which balances loads while accounting for cached prompt behavior.

Centered around the E2 scheduling algorithm, we build *Preble*, a distributed LLM serving system that aims to provide high serving throughput and low request average and tail latency for long and shared prompts. Preble consists of a global, request-level scheduler and a per-GPU, iteration-level scheduler. Apart from E2, Preble incorporates several novel designs to tackle practical LLM challenges. First, with the basic E2 algorithm, a prefix is cached at a GPU after its initial assignment until its eviction. However, the amount of requests sharing it can change over time, which can cause load imbalance. To mitigate this issue, Preble detects load changes and redirects requests from a heavily loaded GPU to a light GPU. If the load hitting a popular prefix increases beyond what a single GPU can handle, Preble automatically scales (autoscales) the prefix by replicating it on multiple GPUs.

Second, prefill and decoding phases have different computation needs, as discovered and tackled by a set of recent works Zhong et al. (2024); Patel et al. (2024); Agrawal et al. (2024). We propose a new way to solve this problem based on our insight that a prompt hitting a cached prefix can be treated as decoding-phase computation, while a missed prompt can be treated as prefill-phase computation because of the high

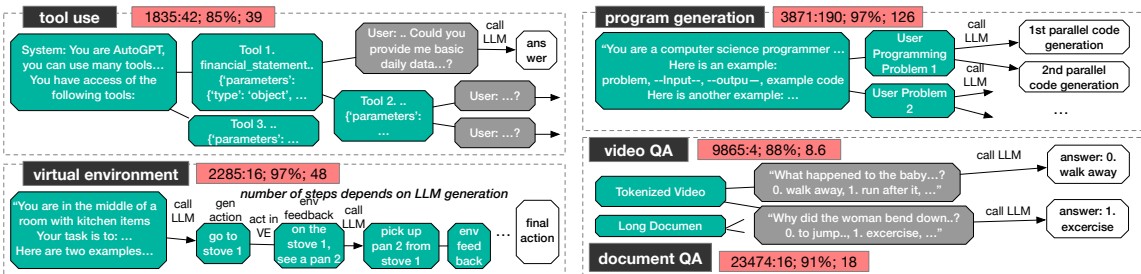

Figure 1: **Prompt Sharing Features of Five Workloads.** *Green boxes represent shared prefixes. Grey boxes are non-shared prompts. White boxes are output generation. Red boxes contain statistics in average values: "prompt-length:output-length; shared token percentage; number of requests sharing a sequence".*

prompt-to-decoding token length ratio. Thus, our global scheduler tries to mix the two types of requests on a GPU to balance prefill and decoding computation needs.

Finally, unlike existing works that either honor request fairness or maximize prefix matching, Preble aims to achieve high prefix reusing while ensuring fairness, which is important in multi-tenancy environments. We achieve this by assigning priorities to waiting requests based on their prefix cache hit ratio and giving each priority their respective quota of requests to serve.

We implement Preble as a standalone layer on top of slightly modified vLLM Kwon et al. (2023) and SGLang Zheng et al. (2023b), two popular open-source LLM serving systems both supporting single-GPU prefix caching. We evaluate Preble using our studied five workloads and the Azure request arrival pattern with the Mistral 7B model Jiang et al. (2023) and the Llama-3 70B model Meta (2024) on a four-Nvidia-A6000 GPU cluster and an eight-Nvidia-H100 GPU server. Our results show that Preble outperforms SGLang by 1.5×-14.5× and 2×-10× on average and p99 average request latency and similarly when compared to vLLM.

Overall, this paper makes the following key contributions: **(1)** The first study of LLM workloads with long and shared prompts, resulting in four key insights. **(2)** E2, a new LLM request scheduling algorithm with the idea of exploitation and exploration integration. **(3)** Preble, the first distributed LLM serving system that targets long and shared prompts. **(4)** A comprehensive evaluation of Preble and SOTA LLM serving systems on two popular open-source LLMs, five real workloads, and two GPU clusters. Preble is publicly available at https://github.com/WukLab/preble.

## 2 Real-World Long and Shared Prompts

This section briefly presents our study results of five LLM use cases and an end-user LLM use trace: tool use Schick et al. (2023), embodied agent in a virtual environment Hao et al. (2023), software program generation Nijkamp et al. (2023), video QA Xiao et al. (2021), long document QA Li et al. (2023a), and the Azure LLM usage trace Patel et al. (2024) (which contains chat and code-generation usages). We pick the five workloads to study and evaluate, as they resemble long-and-shared-prompt use cases reported in production Anthropic (2024). Figure 8 demonstrates the prompt usages and overall features of these workloads. Appendix A presents our full study methodology and results.

Overall, our study shows that prompts are significantly longer than output lengths in the five workloads and the Azure trace, ranging from 4× (Azure chat) to 2494× (video QA). Moreover, prompt sharing ranges from 85% (*i.e.*, 85% prefix tokens in a prompt are shared with at least one more request) to 97% across the five workloads, with embodied agents and program generation having the highest sharing amount. Additionally, a common sequence in a workload is shared by 8.6 to 126 requests on average, with different deviations

across workloads. Finally, the Azure trace indicates high variations in total request loads manifested as different end-user request inter-arrival times that range from 2 microseconds to 217 seconds.

Our study results indicate that LLM serving systems for such workloads should focus on optimizing prefill computation, and a viable way is to cache and share prefixes. However, the variations in prompt-sharing features and request arrival patterns make it challenging to build an efficient distributed serving system.

## 3  Preble Design

We now present the E2 algorithm and the design of Preble, beginning with the overall system architecture of Preble, followed by its global scheduler and local scheduler designs. Preble significantly improves serving speed and reduces request latency for workloads with long and shared prompts. For workloads without any shared prompts, its behavior and performance are the same as SOTA LLM serving systems like vLLM.

### 3.1  Overall System Architecture

Preble is a distributed GPU-based LLM serving system supporting both data parallelism and model parallelism. While its model parallelism support is standard (*e.g.*, tensor parallelism), Preble's scheduling of requests on data-parallel GPUs is designed specifically for long and shared prompts.

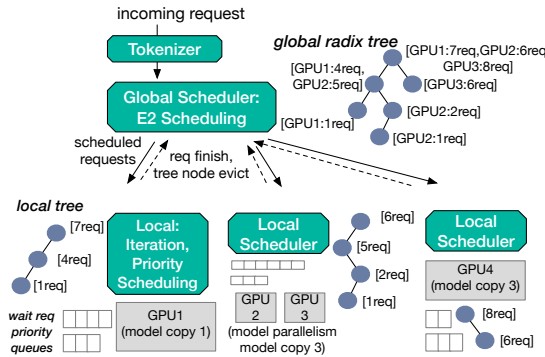

Figure 2: **Preble Architecture.**

We propose a two-level scheduling system where a global scheduler performs *request-level* scheduling decisions and orchestrates the overall load balancing across GPUs, while a per-model-instance local scheduler performs *iteration-level* scheduling for requests assigned to the GPU, as shown in Figure 2. Depending on the GPU cluster topology, the global scheduler may be deployed on a separate server for a multi-server GPU cluster or on the same server as a single-server-multi-GPU cluster. The local scheduler manages one model instance (multiple GPUs when using model parallelism, single GPU when not) and runs on the CPU of the same server as the GPUs. Our current implementation of Preble scales to at least 70 to 391 GPUs. To offer larger, data-center-level scales, one can deploy several Preble clusters, each having one global scheduler.

This design offers several benefits: 1) by having all requests in a cluster go through the global scheduler, we have a centralized place to maintain a global view of cluster load and prompt caching information, both being essential for E2; 2) by performing coarse-grained, request-level scheduling, a single global scheduler can scale to hundreds of GPUs, avoiding the complexity of maintaining multiple distributed global schedulers for a cluster; and 3) by performing fine-grained, iteration-level scheduling at each GPU, the local scheduler can quickly adapt to GPU resource and request availability changes and make precise decisions.

### 3.2  E2 Global Scheduler

We now present our global scheduler design, which centers around the E2 distributed scheduling algorithm.

**Global scheduler data structures.** The global scheduler maintains several data structures to assist its prompt-aware request scheduling. The primary data structure is *global prefix trees*, implemented as radix trees Wikipedia (2024). Each tree has a distinct root storing the shared prefix of all prompts under the tree.

When inserting a new request to the tree, we match its tokens from the beginning (*i.e.*, prefix matching) until no match exists, and we insert the remaining tokens as a new leaf node. If no match exists at all, we create a new tree with this request's prompt as the root node. If an existing tree node only matches partially to the new request (*i.e.*, the prefix of a node matches a sub-sequence of the new request), we split the node into the matched part and the remaining part. For each tree node, we record three pieces of information: the number of tokens in the tree node, the set of GPUs caching the tree node KVs, and the per-GPU number of requests sharing the tree node in a history window $H$. When a tree node has no caching GPU and there is no request within the window $H$ in the whole system sharing it, we remove it from the tree.

**Per-request scheduling policy.** To schedule a request, the global scheduler uses our proposed E2 scheduling algorithm, as illustrated in Algorithm 1. It first matches the request's prompt in the global prefix trees. When the amount of recomputation saved (number of tokens in the matched prefix) is larger than the amount of new computation (number of tokens in the remaining prompt), we favor exploitation over exploration because the gain of saved GPU resources (computation for matched tokens) is higher than the load (computation for unique tokens) that can potentially be balanced in the GPU cluster. For such requests, E2 *exploits* existing cache by assigning the request to the GPU that caches the tree node with the longest tokens in the matched prefix. If multiple such GPUs exist, E2 chooses the GPU with the lightest request load using the load calculation to be introduced next.

If the matched prefix is shorter than the remaining tokens, E2 *explores* the best GPU to run the request based on our proposed new prompt-aware "load cost" definition. Exploration gives E2 a chance to distribute load to different GPUs, which is the key to striking long-term cluster execution efficiency. E2 unifies three types of costs when calculating the per-GPU load: the computation cost aggregated across all requests within a time window, the recomputation cost needed for evicting memory to run the new request, and the computation cost of the new request. E2 calculates all three costs as GPU computation time and finds the GPU with the lowest sum. Instead of profiling the actual computation time, we only maintain token counts at the global scheduler, which largely reduces the system overhead. We leverage transformer-based LLMs' properties that the computation amount of prefill and decoding are proportional to the number of prompt tokens and generated tokens (Figures 9 and 10). Below and in Algorithm 2, we detail the three calculations for scheduling a new request $R_k$.

The first cost is the *overall* GPU computation load $L_i$ for $GPU_i$. We capture a recent load history on $GPU_i$ with a time window $H$ with a default value of 3 minutes (we test different $H$ lengths and find the results insensitive to it). We do not use $GPU_i$'s load at the exact request scheduling time for two reasons: 1) a GPU's load can change between the time of scheduling $R_k$ to the time of running it, and 2) the placement of a prefix has a longer-term effect than a single load in time because of other requests' future exploitation of it. For each request $R_r$ in the history, we estimate its prefill time $PT_r$ with a regression function using the number of tokens in $R_r$ that do not match any prefixes on $GPU_i$; we estimate its decoding time $DT_r$ with another regression function using the average request output length observed on $GPU_i$ in window $H$. We have the first category of load, $L_i = \sum_{r \in W}(PT_r + DT_r)$, where $W$ is the set of requests in the window $H$. The regression functions used in this calculation are captured from offline profiling for each GPU type. Note that even though the number of output tokens is not known a priori, our workload study (Appendix A) shows that it is small and similar across a type of workload. Thus, we use the average output length in $H$ as the estimated decoding length for $DT_r$.

The second cost is the potential cost to free GPU memory so that the new request, $R_k$, can run. Given that GPUs run at full capacity with our and existing serving policies Agrawal et al. (2023), we expect this cost to always occur. Intuitively, the more tokens in a sequence that are shared and by more requests, the more costly it is to evict the sequence. Thus, to calculate the eviction cost, $M_i$, the global scheduler first uses the eviction algorithm to be discussed in Section 3.3 to find the tree nodes on $GPU_i$ that would be evicted to run $R_k$. For each such tree node $j$, its eviction cost is the recomputation time of the evicted tokens multiplied by the hit rate of the node, $N_j$. Thus, we have $M_i = \sum_{j \in E} PT_j \times N_j$ where $E$ is the eviction node set, $PT_j$

is the prefill time for the length of tree node $j$, and $N_j$ is the the number of requests sharing tree node $j$ in history $H$ over the total number of requests on $GPU_i$ in $H$. Note that we do not include the decoding time, as a request's decoding time is unaffected by prefix cache eviction, and decoding costs have already been counted in $L_i$.

The third cost is the actual cost, $P_i$, to run the new request $R_k$ on $GPU_i$, which is simply the prefill time of the missed tokens in request $R_k$. We do not count its decoding time, as it is the same across GPUs, and our goal is to compare the per-GPU load across GPUs.

The total cost of assigning the current request to $GPU_i$ is $L_i + M_i + P_i$ and we choose the GPU with the lowest total cost to assign the request to.

**Post-assignment load adjustment.** With the above algorithm, after the global scheduler assigns a request to a GPU, its prefix lives there until its eviction. This greedy approach works well in cases where the load to a prefix is relatively stable but not otherwise. We propose two ways of managing post-assignment load changes. The first way shifts load between GPUs and is applicable when the load surge can be handled by a single GPU in the cluster. The global scheduler maintains a per-GPU load as discussed above. If the most heavily loaded GPU's load is more than $Th_{bal}$ times higher than the lightest GPU, it shifts load from the former to the latter until their difference is below $Th_{bal}$. $Th_{bal}$ is configurable and can be deducted from profiling GPU and LLM types. To perform this load rebalancing, we direct future requests that are supposed to exploit the heavy GPU to the light GPU.

The second way is to auto-scale a prefix by replicating it and splitting its subtree by load when we detect that a certain prefix's request load is still too high (average queueing time doubles over $H$) even after the above load rebalancing. We calculate the subtree's load using Algorithm 2.

**Prefill-decoding balancing.** From our study results in Appendix A and reported by others, LLM prefill has a larger compute-to-memory ratio than decoding, causing inefficient GPU resource utilization and performance degradation. While various recent works tackle the prefill-decoding imbalance problem by chunking prefills Agrawal et al. (2023) and prefill-decoding disaggregation Zhong et al. (2024); Patel et al. (2024); Strati et al. (2024); Hu et al. (2024); Qin et al. (2024a), we propose a new way of solving the problem leveraging prompt sharing features at a cluster level. Our insight is that a request with its entire prompt shared and cached would only perform the decoding phase. Thus, it can be treated as a decoding-phase computing unit. Meanwhile, a request with a long prompt not cached and a short output length can be treated as a prefill-phase computing unit. A partially cached prompt can be treated as being between the prefill- and decoding-phase units. Thus, we can balance prefill-decoding by combining requests with more or less prompt sharing instead of or in addition to existing balancing techniques.

Specifically, when a request is about to be explored, the global scheduler first considers the prefill and decoding balancing for each GPU. If a GPU is heavily loaded with decoding-phase computing units, the global scheduler directs the current request to it, as a request to be explored will incur recomputation for prompt and is considered a prefill-phase unit. We prioritize this policy over the load-cost comparison (Algorithm 2) because a GPU with heavy decoding has unused computation capacity that we can almost freely use. The global scheduler performs the load-cost comparison if all GPUs have relatively balanced decoding-prefill loads. Apart from this prefill-decoding balancing performed at the global scheduler, our local scheduler also performs traditional chunked prefill for each GPU (Section 3.3).

### 3.3 Local Scheduler

**Local scheduler mechanism.** The local scheduler schedules the requests that the global scheduler assigns to its managed GPU(s). Similar to existing LLM serving systems Yu et al. (2022); Kwon et al. (2023); Zheng et al. (2023b); Aminabadi et al. (2022), we run one local scheduler per GPU and schedule requests at the iteration level. Each local scheduler maintains a request wait queue, a prefix (radix) tree, and the number of active requests sharing each prefix tree node.

When a new request arrives, the local scheduler matches it to the local prefix tree and updates the tree accordingly. It also inserts the request into the waiting queue. After each model iteration, the local scheduler forms the next batch by selecting waiting requests using a priority-based algorithm to be discussed next. If a selected request has a long and non-shared prompt, we chunk the prompt similar to Sarathi Agrawal et al. (2023). If the GPU memory is not enough to run the batch, the local scheduler selects a tree node(s) or part of a tree node (if a part is enough) to evict based on the request accessing time (LRU) of tree nodes. The local scheduler then asynchronously informs the global scheduler about the eviction, and the latter processes it in the background.

**Waiting queue request ordering.** Today's LLM serving systems schedule requests in the waiting queue according to FCFS Kwon et al. (2023) or prefix sharing Zheng et al. (2023b) (serve the request with the highest sharing amount the first). The former ignores prompt sharing and results in more recomputation; the latter ignores fairness and could result in starvation Wu et al. (2023). We propose a priority-based wait queue scheduling policy that considers both prefix sharing and fairness. Specifically, we create $P$ (a configurable parameter) priority groups and assign a request to the priority group according to its cached token percentage. For example, if 63 out of 100 tokens in a request's prompt are cached on the GPU and $P$ is 10, it will be assigned priority six. When selecting requests to form the next batch, the scheduler proportionally selects requests from each priority group, with the higher priority group getting more requests selected than lower priority ones. For example, if 55 requests are to be selected to form a batch, the scheduler picks ten from priority group 10, nine from priority 9, etc.

## 4 Implementation and Evaluation Results

### 4.1 Implementation

We implemented Preble as a standalone layer to perform distributed LLM serving. As such, Preble can be added to any existing serving systems with no or minimal changes — we currently support vLLM Kwon et al. (2023) and SGLang Zheng et al. (2023b) as two backends.

**Global scheduler scalability.** In implementing the global scheduler, we use a few techniques to improve its scalability. Incoming requests are first tokenized by a parallel tokenization layer. Afterward, the global scheduler spawns asynchronous request handlers to schedule requests. Access to the global radix tree during request handling is lock-free, as most operations are read-only. The only exceptions are updating a GPU to be assigned to a tree node and the increment of request count hitting the tree node, both of which can be expressed as atomic instructions. Additionally, the global scheduler maintains a current load count for each GPU by updating it every time a new request is assigned to it or when it evicts a tree node. Thus, our realization of the E2 algorithm is performance-efficient. Finally, to ensure foreground request performance, the global scheduler runs non-request-scheduling tasks such as rebalancing and eviction bookkeeping in the background with separate threads.

### 4.2 Workloads and Environments

**Workloads.** We evaluate our results on five LLM use cases: LLM generation with tool demonstration and calling Schick et al. (2023), LLM interacting with virtual environments as an embodied agent Hao et al. (2023), LLM for program generation Nijkamp et al. (2023), answering questions about videos Xiao et al. (2021), and answer questions about long documents Li et al. (2023a). Their properties are in Appendix A.

For each workload, we sample enough requests to fulfill the request-per-second (RPS) needs and GPU setup (*e.g.*, a larger GPU or more GPUs can handle more). For experiments other than the ones using the Azure Inference Trace, we set the inter-arrival time using a Poisson distribution with a mean that around the RPS (X-axis in most figures). We then run the experiments until stable state is reached and for a significant length.

**LLMs and environments.** We test Preble and baselines using two popular open-source LLMs, the Mistral 7B model Jiang et al. (2023) and the Llama-3 70B model Meta (2024). We run our experiments in one of the two environments: a two-server cluster with two NVidia A600 GPUs and one eight NVidia-H100-GPU.

**Baseline.** Our baselines are serving systems that support single-GPU prefix sharing, including SGLang Zheng et al. (2023b) and vLLM (which recently added a beta feature for prefix sharing Moore & Li (2024)). To run SGLang and vLLM in a distributed fashion, we set up a load balancer that sends requests in a round-robin fashion to individual SGLang/vLLM instances (*i.e.*, non-prompt-aware data parallelism). As round-robin distributes requests evenly, these baselines capture a distributed serving system that balances request loads and then performs prefix sharing within each parallel instance.

**Metrics.** We use three key metrics: request per second, which measures serving capacity; average end-to-end request latency (including scheduling time, queueing time, prefill, and decoding time); and p99 request latency. Note that our metrics differ slightly from some existing LLM serving works Kwon et al. (2023); Yu et al. (2022), as we do not use TPOT (time per output token) or TTFT (time to first token) as key metrics. This is because our target LLM use has short output lengths, rendering TPOT not as meaningful and TTFT close to the request latency. We consider p99 latency since it is important to control the tail latency in LLM serving as with all other user-facing services DeCandia et al. (2007); Ongaro et al. (2011).

### 4.3 End-to-End Workload Performance

We first present the overall performance of Preble and the baselines. Below, we focus on the comparison with SOTA SGLang as it is specifically designed for (single-GPU) prefix sharing. We provide Preble's comparison to vLLM and to different SGLang versions in the Appendix D.

**Single workload results.** We now present the average and p99 latency against increasing requests arriving per second (RPS) of Preble and SGLang on the five workloads, two LLMs, and two GPU environments, as shown in Figure 3. Overall, Preble significantly outperforms the data-parallel SGLang baseline for all settings, as can be seen from Preble's lower average and p99 latency, especially under higher RPS (or the other way around, for the same latency target, Preble can serve higher RPS). Our improvements over SGLang range from $1.5\times$ to $14.5\times$ in terms of average latency and $2\times$ to $10\times$ in p99 latency.

Comparing across workloads, we see bigger improvements of Preble over SGLang on the Toolbench, embodied agent, video QA, and LooGLE workloads than the programming workloads. The programming workload has the longest decoding length among all the workloads. As decoding time starts to dominate total request latency, and we do not improve decoding performance, the room for improvement for Preble is smaller. Nonetheless, Preble still achieves $1.56\times$ to $1.8\times$ improvement in average latency and $3\times$ to $4\times$ in p99 latency over SGLang in the programming workload.

Comparing across the number of GPUs, Preble's relative improvement over the baselines stays similar when going from 2 to 4 A6000 GPUs. Considering absolute values, we see Preble successfully maintain similar latency even as RPS doubles, showing its strong scalability. When changing from A6000 to 8 H100 and switching the Mistral 7B to Llama-3 70B, we find relative improvements of Preble to increase.

**Azure trace and mixed workloads.** Our experiments above use a Poisson request arrival distribution (which is the same as most existing LLM works' experimental methodology Kwon et al. (2023); Li et al. (2023b)). To understand Preble's performance under real-world request load, we run the tool use and video QA workloads using Azure's LLM request arrival pattern (Appendix A.6) instead of Poisson distributions. Here, we mix the two workloads to mimic Azure's mixed chat and code traces. As shown in Figure 4, Preble has significant improvements in average and p99 latencies and on average TTFT and TPOT.

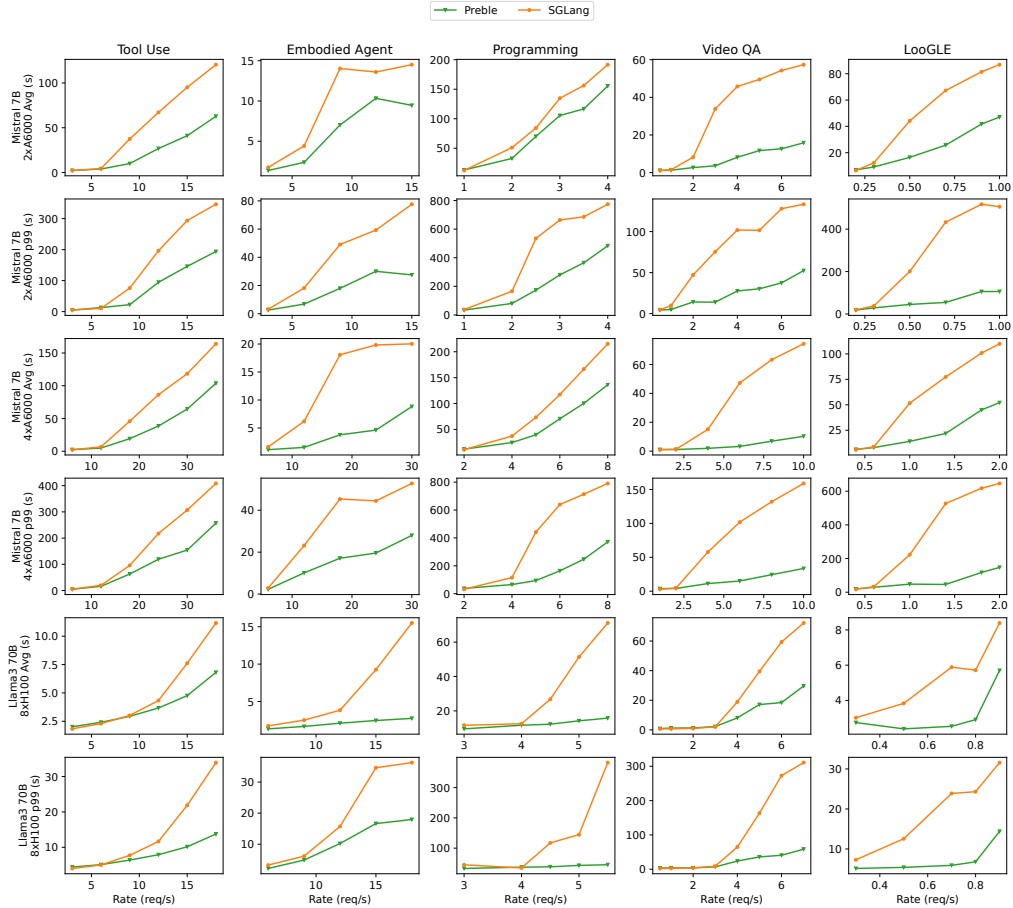

Figure 3: **End-to-end Workload Performance** *The top and middle two rows run on 2-4 A6000s with Mistral 7B. The bottom two rows run on 8 H100s set up as 4-GPU tensor parallelism plus data parallelism with Llama-3 70B.*

## 4.4 Deep Dive

We now provide a detailed analysis of Preble, including an ablation study and global scheduler scalability test. Because of H100 GPUs' high cost and low availability, we run all experiments in below with A6000s.

**Ablation study.** To understand where the benefits of Preble come from, we evaluate Preble by incrementally adding features presented in Section 3. We chose the tool use workload with a Zipf-1.1 popularity distribution among the prompts in the dataset to represent real-life skewed tool popularity. Other workloads and distributions benefit from a different set of techniques. We start with using the SGLang round-robin baseline. We first add the per-request E2 policy (Section 3.2), which results in an improvement on both average and p99 request latency because of E2's dynamic load partitioning. We then add the post-assignment global rebalancing and autoscaling, which successfully balances out load even more, resulting in further improvement, especially with p99. Further adding the prefill/decode-aware handling results in more improvement on both average and p99, since it considers the current batch composition and is able to better utilize the GPU resources. Finally, we add the local-scheduler priority-based wait-queue scheduling (§3.3), which, as expected, improves p99 but not average latency, as its goal is fairness.

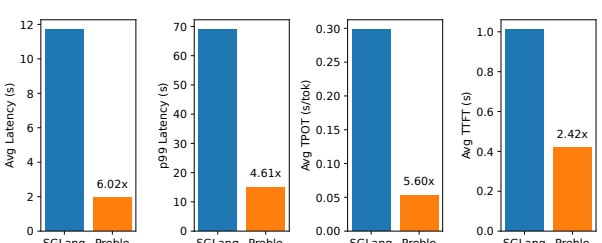
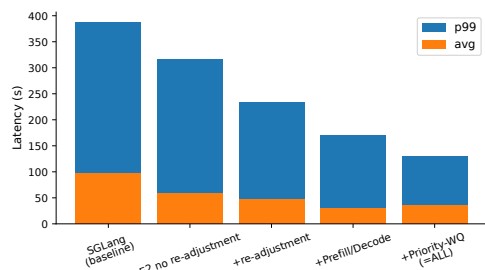

Figure 4: **Mixed Workload With Azure Trace** *A mix of Tool and Video workloads with Azure arrival trace on 4 A6000s.*

Figure 5: **Ablation Results** *Running ToolBench with Zipf-1.1 running on 4 A6000s*

**Global scheduler performance and scalability.** We measure Preble's global scheduler max throughput by sending large (e.g., 50,000) requests at once, eliminating arrival pattern effects and ensuring saturation. Since the global prefix tree search is the most time-consuming task at the global scheduler, we test the Toolbench and VideoQA workloads, which have the most complex and simplest prefix tree structures in our five workloads. Preble's global scheduler achieves a processing rate of 245 and 2931 requests per second for Toolbench and VideoQA. We also measure the network processing speed and find it not to be the bottleneck. With the peak GPU processing rate (30-150 tokens per second decoding speed with Mistral 7B on A100) and our workloads' output length (Table 1), one Preble global scheduler can sustain at least 70 to 391 concurrent A100s. If accounting for prefill time or bigger models, our scheduler would sustain even more GPUs.

## 5 Related Works

LLMs' usages are shifting to be more prompt-heavy. As a result, the problem of prefill and decoding having different compute-to-memory ratios is exacerbated. Several recent works have targeted LLM usages with long prompts and proposed solutions to solve the imbalance problem. The first approach, called *chunked prefill*, chunks a prompt and runs each chunk with other decoding requests in a batch in a single iteration to reduce waiting Agrawal et al. (2023; 2024). The second approach is to separate prefill and decoding to different GPUs to avoid prefill-decoding interference Patel et al. (2024); Zhong et al. (2024). These solutions target long prompts but ignore prompt sharing. Preble consider prompt length and sharing, and we use a novel sharing-based approach on top of chunked prefill to solve the prefill-decoding imbalance problem.

Many state-of-art LLM inference systems Zheng et al. (2023b); Moore & Li (2024) enable prefix sharing to improve KV cache reuse. However, these systems limit the sharing at the scale of individual model instance, overlooking the cluster-level prefix-sharing opportunities and challenges. More recent works Hu et al. (2024); Qin et al. (2025) corroborate our insight in adding prompt-sharing awareness to the cluster-level scheduler. Unlike Preble, MemServe Hu et al. (2024) prioritizes cache reuse regardless of current loads and thus could result in imbalanced and inefficient GPU usage. Mooncake Qin et al. (2025) is a concurrent work of Preble and similarly balances cache exploitation and load distribution. Differently, Preble factors the potential KV eviction in the load estimation, which can be more accurate. Moreover, Preble also improves fairness in prefix-caching with better waiting request ordering.

## 6 Conclusion

This paper identified the problem of distributed serving for long and sharing prompts. To solve this problem, we performed a study on five LLM workloads and one real LLM trace. We presented E2, a distributed LLM request scheduling algorithm targeting LLM usages with long and shared prompts. We built Preble, a distributed LLM serving system on top of the E2 algorithm and a hierarchical scheduling architecture. Our results show that Preble significantly improves LLM serving performance over SOTA serving systems while ensuring request fairness and controlling the tail latency.

# Acknowledgment

We would like to thank the anonymous reviewers for their tremendous feedback and comments, which have substantially improved the content and presentation of this paper. We are also thankful to Yufei Ding and Geoff Voelker for their support and feedback of this work. This material is based upon work supported by gifts from AWS, Google, and Meta. Any opinions, findings, conclusions, or recommendations expressed in this material are those of the authors and do not necessarily reflect the views of these institutions.

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

# 7  Appendix

# A  Study on LLM Prompts

Today's LLM usage goes beyond simple chatting. As LLM usage becomes more commercialized, LLM prompts become more structured and complex, outshadowing the text an LLM generates. This section presents our study results of five popular new LLM use cases: tool (or API, agent) use Schick et al. (2023), interacting with virtual environments as an embodied agent Hao et al. (2023); Huang et al. (2022), software program generation Nijkamp et al. (2023), answering questions about videos Xiao et al. (2021), and answer questions about long documents Li et al. (2023a). Figure 8 demonstrates the prompt usages of these workloads. We study each case with real public datasets and understand their prompt features from a systems perspective. For datasets that do not provide outputs, we use Llama-3 7B model as the LLM to generate outputs. For each dataset, we construct a prefix tree for all the requests in the dataset (*i.e.*, assuming an infinite prefix cache).

Table 1 and Figure 6 summarize our study results, including prompt and decoding (output) length, amount of sharing in a prompt, key portion size in a prompt, and number of requests sharing a key portion. We define the "key portion" of a request as the deepest node in a path that has more tokens than the sum of its predecessors.

To understand real-world LLM user request features, we study a recently released public cloud LLM trace. This section ends with our summary insights.

| Workload | Prompt Len | Output Len | Shared Prefix in Prompt | KeyPort. in Prompt | Req Share KeyPort. |
|---|---|---|---|---|---|
| Toolbench | (1835, 742) | (43, 16) | (85%, 13%) | (76%, 16%) | (39, 64) |
| Embodied Agent | (2285, 471) | (16, 13) | (97%, 14%) | (76%, 12%) | (48, 8) |
| Programming | (3871, 1656) | (190, 343) | (97%, 7.4%) | (78%, 13%) | (126, 2157) |
| Video QA | (9865, 5976) | (4, 1.5) | (88%, 32%) | (99%, 0.2%) | (8.6, 2) |
| LooGLE | (23474, 6105) | (16, 9.9) | (91%, 24%) | (94%, 15%) | (18, 8.6) |

Table 1: **LLM Prompt Properties** *Each cell except for number of requests shows (mean, standard deviation). Length represented using number of tokens. "KeyPort." stands for Key Portion.*

## A.1  Tool Use

Today, LLMs are often augmented by various tools such as calculators and web searches. To equip a model with the ability to invoke a tool, it must be given the correct syntax for querying the tool, along with examples (or "demonstrations") of tool use. We evaluate the Toolbench Guo et al. (2024) dataset, which consists of more than 210k queries that call over 16k unique tools. Each query shares the same system prompt followed by tool-specific instructions. The final part of the query is the user's specific question or task. These are all concatenated together to form the final prompt. We find that most of the sharing comes from queries that all share the same tool, and these instructions can be 43x longer than the output length. The Toolbench workload is also representative of other tasks that "prep" an LLM in a similar fashion. For example, instead of tool-calling, LLMs can have roles layered on top of the system prompt, which is popular in emerging systems that utilize the same LLM with multiple roles to create an ensemble Wu et al. (2024); Li et al. (2024b;a).

## A.2  Embodied Agents

LLMs are increasingly found in agents that can interact with environments, such as a player in a role-playing game or controlling a robot. In this scenario, the LLM receives feedback from the environment, forms an action, and then "performs" the action. This is conducted in a loop until the model has achieved the goal.

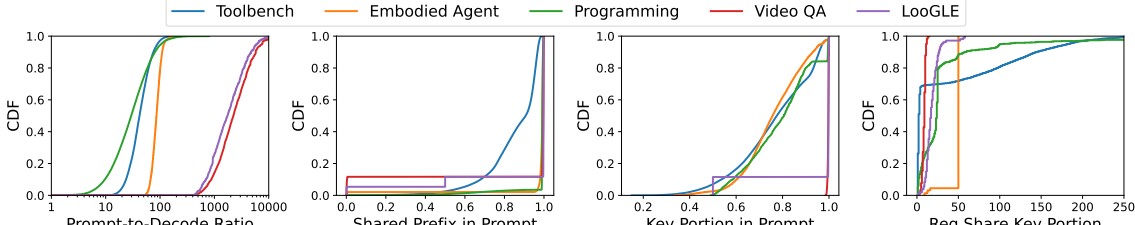

Figure 6: **CDF Plot of Key Metrics** *Showing CDF for all five workloads on prompt-to-decode ratio, shared prefix*

The workload we utilize is sourced from the ALFWorld Shridhar et al. (2021) dataset and has 7.5k requests. Prompts first describe the environment and the task, followed by a demonstration of steps to solve the task. The model then solves its given task by looping over a planning step followed by an action step. After each action, the text-based environment returns an observation that the model incorporates into its next planning step. Every new invocation to the LLM in this loop is treated as a new request, resulting in each step sharing the context of previous steps. Interestingly, the number of steps is determined by LLM generation, creating an unpredictable sharing pattern. Because steps are chained together, prompts are still 157x longer than output tokens.

The embodied agent workload can represent a wide variety of other use cases, such as chain of thought Yao et al. (2023b); Wei et al. (2024), multi-turn tool usage Wang et al. (2024); Qin et al. (2024b), and chatbots Zheng et al. (2023a). Any dependency between the model and the outside environment can be considered an agent receiving feedback.

### A.3    Program Generation

One of the popular uses of LLMs is to generate software programs Nijkamp et al. (2023). We study the APPS competitive programming dataset Hendrycks et al. (2021), a dataset of programming problems. To generate better-quality programs, an approach taken by a recent paper Juravsky et al. (2024) is to add a demonstration of several generic code examples before the user problem to instruct an LLM. This added demonstration is the same across all problems and becomes the system prompt. Following the system prompt is the programming problem description. Afterward, this approach invokes the LLM several times in parallel to generate multiple candidate programs, out of which the best is chosen to return to the user. As generated code is relatively long (compared to outputs of other workloads we study), the prompt-to-output ratio (20x) is relatively low. Prompt sharing comes from two places: the system prompt of code demonstration is shared across all requests, and the programming problem is shared across all parallel generations. Depending on how complex the problem is, its description could be longer or shorter than the system prompt; a problem description can also be partially the same as another problem description. Such complexity results in competitive programming having diverse key-portion properties. Such example demonstration and parallel generation technique is common in recent prompt engineering, for example, with ReAct Yao et al. (2023b), Tree-of-Thoughts Yao et al. (2023a), and Self Consistency Wang et al. (2023).

### A.4    Video Question and Answer

The advent of video models like OpenAI Sora OpenAI (2024b) has created an explosion of interest in multi-modal models. The use of LLMs, then, goes beyond natural language. A recent usage is to answer questions about videos by tokenizing a video segment and inputting it to an LLM Yu et al. (2023); Esser et al. (2020). To study this, we analyze the NExT-QA benchmark Xiao et al. (2021), which consists of 8.5K questions for 1000 video segments. Prompts to the LLM consist of a tokenized video followed by a multiple-choice question. Because of the multiple-choice nature, the outputs of this dataset only have six tokens. Long tokens for representing videos plus short outputs result in this dataset having the highest prompt-to-decoding token

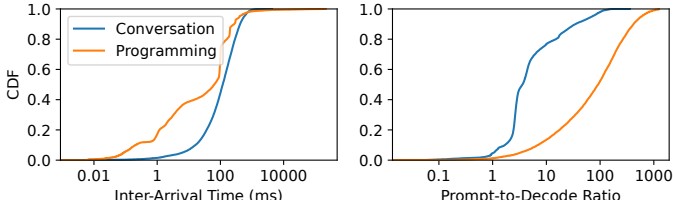

Figure 7: **Azure LLM Trace Analysis Results.**

ratio of all workloads we explored, with nearly 2500× more prompt tokens. Apart from videos, images and audio can also be tokenized to have LLMs answer questions, and we expect them to have similar properties as video QA.

### A.5    Long Document Question and Answer

With newer models, the maximum context length has increased substantially Munkhdalai et al. (2024); Liu et al. (2024); Jacobs et al. (2023), with the latest development supporting 1M tokens Munkhdalai et al. (2024). Longer contexts enable new LLM applications such as asking questions about a long document or even a book. We evaluate this usage with the LooGLE dataset Li et al. (2023a), a collection of 776 long documents and over 6.4k questions. LooGLE has a small system prompt of 13 tokens followed by a long document and then a question about the document. As a common practice, a user or multiple users often ask multiple questions to the same document, resulting in large amounts of shared tokens. Meanwhile, the answers are usually short (*e.g.*, a true or false). These features result in high prompt-to-decode ratio and high sharing ratio in LooGLE.

### A.6    LLM Usages in the Wild

To understand LLM usage in the wild, we analyze the recently released Azure LLM Inference Trace Patel et al. (2024). The trace includes two types of LLM usages: program generation and chat conversation. It provides request arrival time, prompt length, and decode length. As it does not provide actual request content, it is not feasible for us to evaluate prompt content or sharing. Figure 7 plot our analysis results in CDF. We find that the arrival rate is approximately 5 requests per second for chat conversation and 7 requests per second for programming. On average, chat requests arrive 118 ms apart while programming requests arrive 63 ms apart. The mean prompt-to-decode ratio for chat conversations is 4. Since we have no details about shared context from follow-up conversations, this number is expected to be much lower. For the longest 20% of all chat prompts, the mean prompt-to-decode ratio is 175, which is consistent with our observations on other workloads.   For programming, the mean prompt-to-decode ratio is 92 for all prompts. This falls within the range of all the workloads we evaluated.

### A.7    Summary Insights

Our analysis of the five real-world LLM workloads and a real user LLM request trace reveals several key findings.

**Insight 1:** Contrary to popular belief, prompts are significantly longer than output lengths because LLMs support longer context and new LLM usages keep emerging. We believe this trend will continue as LLMs are augmented with more capabilities. **Implication 1:** Optimizing prefill computation can largely improve overall application performance, and imbalanced prefill and decoding computation features should be considered in LLM serving.

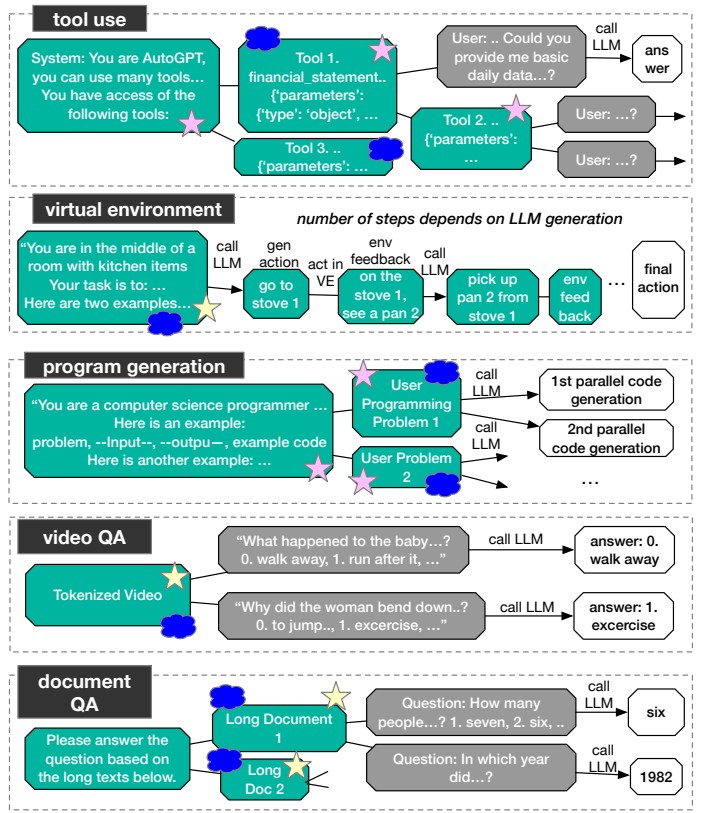

Figure 8: **Workload Demonstration.** *Green boxes represent shared prefixes. Grey boxes are non-shared prompts. White boxes are output generation. Yellow star represents key portions that always happen at fixed parts; pink stars at non-fixed parts. Blue clouds represent the parts that would be used for distributing prefixes if knowing the oracle.*

**Insight 2:** Prompt sharing, or reuse, is common, and the sharing amount is high. Sharing can come from different user requests needing the same tools or instructions to solve a task. It can come from a user asking multiple questions about the same document or video. Context sharing can also happen within the same user task that is solved with a chain or a tree of steps. **Implication 2:** Reuse computation across shared prefixes can largely improve real workloads' performance and should be efficiently supported by distributed LLM serving systems.

**Insight 3:** Most requests have a portion of the prompt sequence that gets a different degree of sharing and is longer than its prefix, reflected as a key portion in prefix trees. Key portions account for the majority of prompts and are shared by a significant amount of requests. **Implication 3:** Identifying the key portion of prompts and optimizing the placement of requests according to their key portions is a viable way of reducing the complexity of scheduling while achieving good performance.

**Insight 4:** Real-world LLM usages have varying load intensity, and different usages (programming vs. conversation) have different loads. Real-world prompts are also much longer than decoding length, but different usages have different prompt-to-decode ratios. Still, the longest prompts are significantly longer. **Implication 4:** An efficient LLM serving system should consider complex, mixed-usage scenarios and factor in both load and prompt sharing variations.

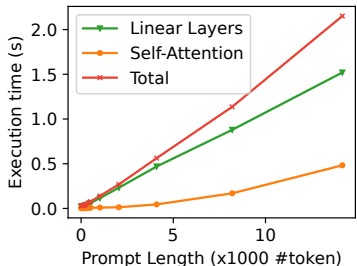

Figure 9: **Prefill Time Decomposition**

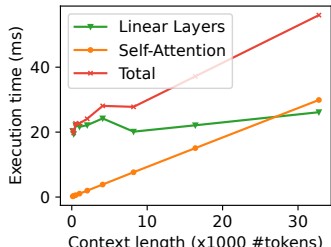

Figure 10: **Decoding Time**

## B  Prefill/Decoding Times

The prefill and decoding stages exhibit different computation behaviors, with the former being computation-bound and the latter being memory-bandwidth bounded. To understand their behaviors and to acquire prefill/decoding computation time functions to be used by E2, we profile the prefill and decoding stage performance with Mistral 7B on the A6000 GPU. Figure 9 plots the prefill time and its breaking downs when prompt length increases. As seen, longer prompts increase prefill time, suggesting that the more savings we can get from prefix sharing, the lower prefill time will be. Moreover, since the linear layer dominates the model forwarding at the prefill stage, the prefill time is overall linear to the prompt length. Figure 10 shows the performance of a single request's decoding performance with varying context lengths (the length of the prompt sequence plus the sequence generated thus far). We observe a similar linear relationship to context token length. Overall, these profiling results suggest that attention computation is regular. Thus, we could use the token length with a profile regression function to estimate computation time.

## C  Algorithms

Below, we provide the global and local scheduling algorthim used by Preble.

---

**Algorithm 1** E2 Global Scheduling Algorithm

---

    **function** SCHEDULEREQUEST($R_k$)
        Match $R_k$ to global radix tree
        $cached\_len \leftarrow$ sum of matched length
        $missed\_len \leftarrow prompt\_len - cached\_len$

        **if** $missed\_len < cached\_len$ **then**            ▷ Exploit $R_k$
            $K \leftarrow$ GPUs with longest node in matched path
            **for each** GPU $i$ in $K$ **do**
                $Cost_i \leftarrow$ LOADCOST($i, R_k$)
            **end for**
            **return** $i$ with lowest $Cost_i$
        **else**                           ▷ Explore $R_k$
            **for each** GPU $i$ in all GPUs **do**
                $Ratio_i \leftarrow$ DECODERATIO($i$)
            **end for**
            ▷ IMBALR: calc based on GPU type and LLM
            **if** highest $Ratio_{max} >$ IMBALR **then**
                **return** $max$
            **end if**

            **for each** GPU $i$ in all GPUs **do**
                $Cost_i \leftarrow$ LOADCOST($i, R_k$)
            **end for**
            **return** $i$ with lowest $Cost_i$
        **end if**
    **end function**

---

---

**Algorithm 2** GPU Load Cost Calculation

---

$\triangleright$ Load cost calculation for GPU $i$ and request $R_k$
**function** LOADCOST($i$, $R_k$)
    $L \leftarrow 0$; $M \leftarrow 0$; $P \leftarrow 0$;

    $\triangleright$ Calculate total load on GPU $i$
    **for each** $R_j$ in history $H$ **do**
        $missed\_len \leftarrow$ non-cached prompt length for $j$
        $L \leftarrow L + \text{PREFILLTIME}(missed\_len)$
        $decode\_len \leftarrow$ average request output length in $H$
        $L \leftarrow L + \text{DECODETIME}(decode\_len)$
    **end for**

    $\triangleright$ Calculate eviction cost
    $E \leftarrow$ tree nodes to evict on GPU $i$ to run $R_k$
    **for each** $j$ in $E$ **do**
        $N_j \leftarrow$ number of requests sharing $j$ in $H$ / number of total requests in $H$
        $M \leftarrow M + \text{PREFILLTIME}(\text{length of } j) \times N_j$
    **end for**

    $\triangleright$ Calculate cost to run $R_k$
    $missed\_len\_k \leftarrow$ non-cached prompt length for $R_k$
    $P \leftarrow \text{PREFILLTIME}(missed\_len\_k)$

    **return** $L + M + P$
**end function**

---

**Algorithm 3** E2 Local Scheduling Algorithm

---

**function** SCHEDULEREQUESTS
    **for all** requests in waiting queue **do**
        Determine prefix hit ratio
        Assign to priority group based on hit ratio
    **end for**
    Process groups in round-robin fashion with limits
**end function**

---

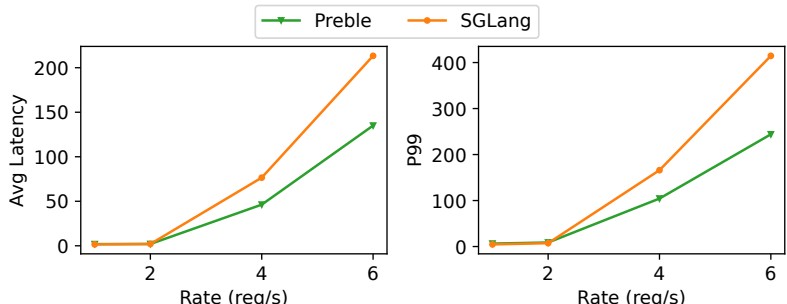

Figure 11: **vLLM Backend Performance** *Evaluated on the Video QA workload using the Mistral 7B model on 2 GPUs.*

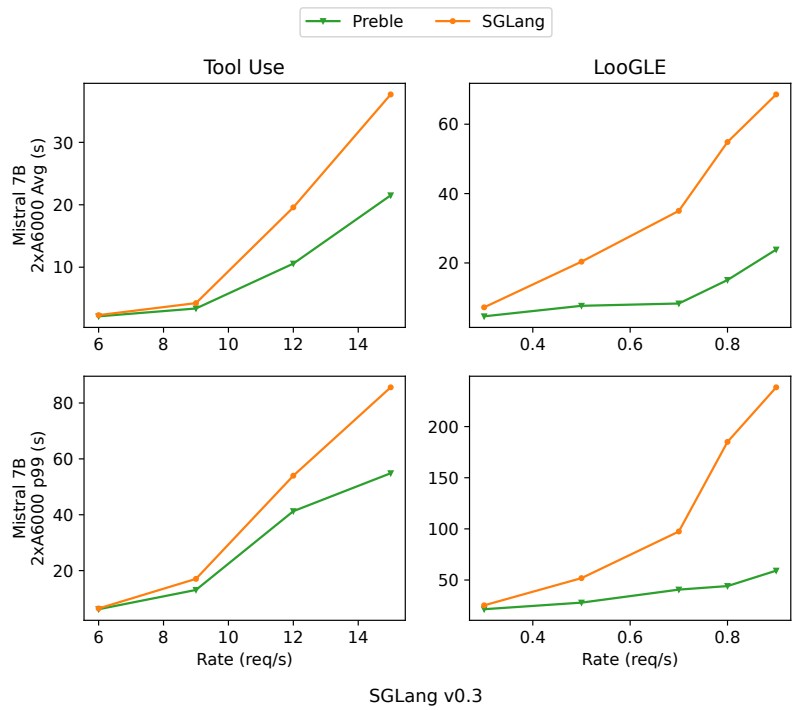

Figure 12: **Latest SGLang v0.3 and Flashinfer 0.1.6**

# D    Comparison with vLLM and Other SGLang Versions

To demonstrate Preble's versatility with multiple LLM backends, we evaluate Preble on vLLM with the vanilla vLLM as the baseline. vLLM recently added support for prefix caching, which we include in the baseline. We use a slightly different version of the Mistral 7B model (v0.2) for this experiment, as vLLM only supports this version. Figure 11 plots the results of running the VideoQA workload on 2 GPUs and the Mistral 7B v0.2 model for both Preble and vLLM. Compared to SGLang as a backend, vLLM as a backend

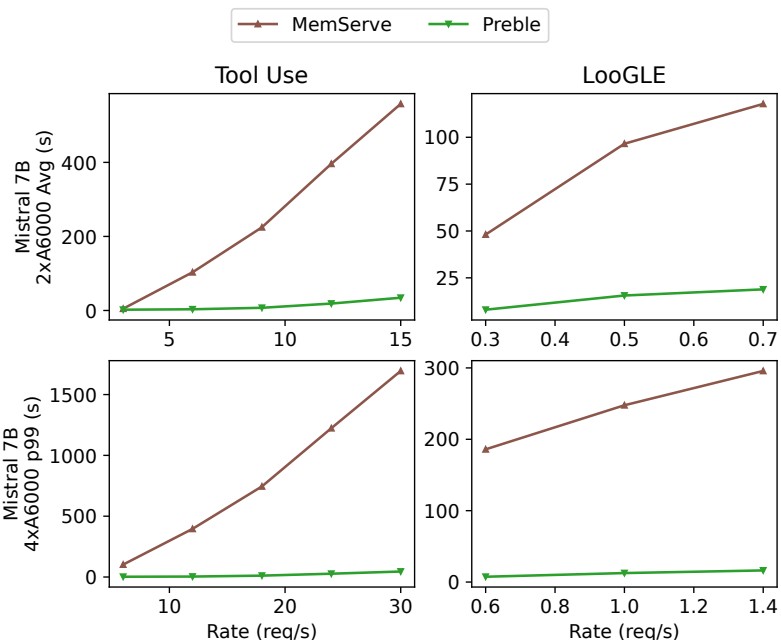

Figure 13: **Compare with MemServe** *Evaluated on the Tool Use and Video QA workloads using the Mistral 7B model with TP=1.*

gives Preble less relative improvement for several reasons: 1) local-GPU prefix sharing is in beta version and not as performant as SGLang; 2) vLLM does not use the flash_infer kernel which makes prefix sharing more efficient; and 3) vLLM does not support chunked prefill together with prefix caching. 4) vLLM has significant scheduling delay

To demonstrate Preble's versatility across multiple SGLang versions, we evaluate the latest SGLang version (v0.3) Team (2024b) and the latest FlashInfer kernel (v1.6) Team (2024a) in addition to the v0.1.12 we used in Section 4. SGLang made two major changes between these two versions: 1) the new FlashInfer kernel improved the performance of sharing 32K context length in the same batch, and 2) SGLang reduced its scheduling overhead and applied other engine optimizations. Figure 12 plots the results of running LooGLE and Toolbench with SGLang v0.3 and Preble. Overall, Preble's improvements over SGLang persist across SGLang versions.

# E    Comparison with Memserve

We also provide a comparison against MemServe, a distributed serving system with prefix awareness, in Figure 13. Preble achieved significantly better average latency in all scenarios. This is due to the over-exploitation of prompt sharing in MemServe's scheduler policy as they route requests based on the longest-prefix-match regardless of the existing load on that node.

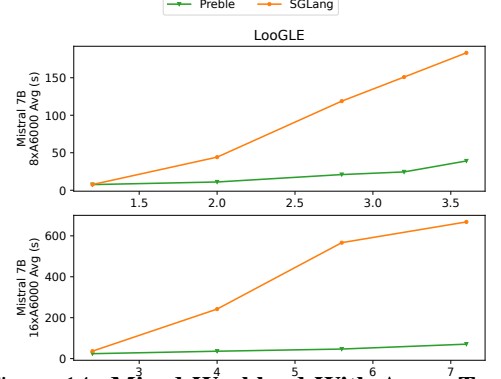

Figure 14: **Mixed Workload With Azure Trace** *A mix of Tool and Video workloads with Azure arrival trace on 4 A6000s.*

Figure 15: **Ablation Results** *Running ToolBench with Zipf-1.1 running on 4 A6000s*

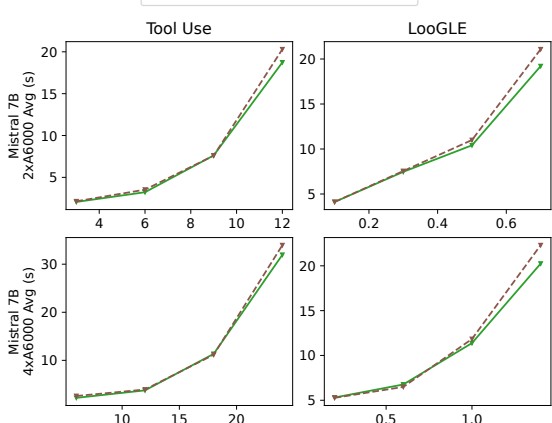

Figure 16: **GPU Utilization Based Dynamic E2 Algorithm**

## F   Scheduler Scalability

As the number of GPUs and unique prefixes increases, the global prefix tree (and thus the amount of work that the global scheduler needs to handle) also grows. To verify Preble's scalability, we test the system using an 8-GPU and a 16-GPU cluster with proportionally more unique prefixes from the LooGLE workload. We provide the average latency in Figure 14. We also show the max sustainable RPS with the 20-second latency requirement of Preble and Round Robin baseline in Figure 15. Based on the results, we can see that there's a linear scalability as instances increase.

## G   Dynamic E2 Scheduling

We also provide an experiment with dynamically selecting exploitation vs exploration based on the total load of the cluster as well as token count in Figure 16. The results between the two approaches are relatively similar. In LooGLE dataset, the default Preble performs (min: -4%, max: 9.2%, avg: 3%) better than the knob based Preble. In the Toolbench dataset, the default Prebleperforms (min: -1.6%, max: 15%, avg: 5.43%) than the dynamic knob based Preble.

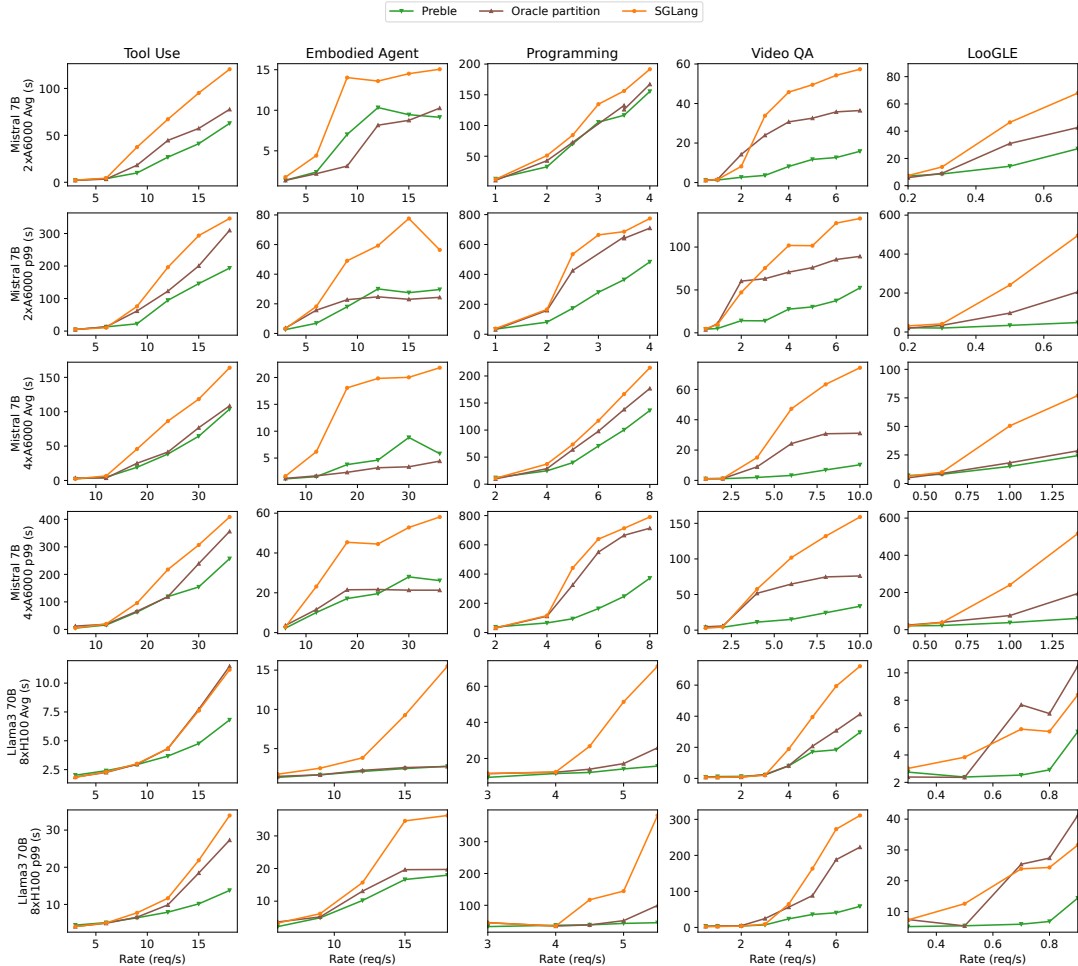

Figure 17: **End-to-end Workload Performance** *The top and middle two rows run on 2-4 A6000s with Mistral 7B. The bottom two rows run on 8 H100s set up as 4-GPU tensor parallelism plus data parallelism with Llama-3 70B.*

## H   Oracle Baseline

We provide an Oracle Partitioning baseline inspired by consistent hashing. Unlike partitioning requests, another way of distributing load is to partition state (i.e., prefix trees in our case). As prefix trees are dynamic, and perfect partitioning requires solving an NP-hard problem, we use an oracle approach to construct this baseline. Specifically, at offline time, we combine all requests in a workload in a single prefix tree, manually examine the key-portion tree layer, and partition the tree into K (number of data-parallel instances) subtrees by evenly splitting this layer (e.g., with the hash values of tokens). For certain workloads, no single layer is clearly or always the key portion; we then partition the tree according to the top layer that can be a key portion. The blue clouds in Figure 8 mark the tree node layers that we manually choose as the partitioning layer for this oracle baseline. Note that such oracle information cannot be acquired in a real serving system. After a tree has been partitioned, we let the online serving system consistently send requests hitting a partition to the corresponding GPU

From the results in 17, we can see that Preble is better or on par with the oracle partition baseline for all setups. This is because key portions cannot always be easily identified and partitioned statically and because even when a tree can be evenly partitioned by size, requests hitting each partition can be different and can change over time.

Preble has more improvements over the Oracle baseline on the VideoQA and Programming workloads than the other three workloads. The VideoQA workload has more questions for longer videos, causing non-even distribution of request load across videos. Even though the Oracle can evenly split videos, the load is not evenly split. The Programming workload's Oracle split is at the layer of user problem, which is not a good, stable indicator of the key portion or the request load. Thus, for these two workloads, Oracle performs much worse than Preble. On the other hand, the embodied agent workload's key portion is the initial instruction, and each request has a different instruction. Thus, by splitting the initial instruction, Oracle can almost perfectly split the load and cache prefixes. The Tool workload has a relatively uniform request distribution across different tool uses, allowing Oracle to partition load evenly.

