# OpenReview forum: "Preble: Efficient Distributed Prompt Scheduling for LLM Serving"
_ICLR.cc/2025/Conference — ICLR 2025 Poster_

### Official Review · Reviewer_NYyJ · 2024-11-01

**Soundness:** 3
**Presentation:** 3
**Contribution:** 3
**Rating:** 8
**Confidence:** 3

**Summary:**

In some applications of LLMs, prompts can be much longer than decode length and could be potentially shared across requests.
However, existing works on KV cache sharing only target a single GPU setting, lacking a cache-aware system for distributed LLM serving.
This paper proposes Preble, a distributed LLM serving platform that utilizes prompt sharing efficiently.
The core of Preble is a two-level scheduler containing a global request scheduler and a local iteration scheduler.
The global scheduler schedules requests based on exploitation and exploration, trading off between load balancing and prefix sharing.
The local iteration scheduler performs priority-based scheduling to ensure fairness.
Evaluations show that Preble can improve greatly on average and p99 request latency compared to SOTA systems.

**Strengths:**

1. Good observation that the prefix sharing ratio is high among popular LLM applications.
2. The two-level scheduling algorithms overall make sense.
3. The evaluation results show solid improvement in request latency compared to SOTA systems.

**Weaknesses:**

1. Lack of justification for some scheduling decisions made in the paper.
2. The workload may not reflect the real distributed LLM serving scenario.

**Questions:**

Thank you for submitting the work to ICLR 2025! I enjoy reading it.
I think the problem statement is clear, the design and implementation of the scheduler make sense on a high level, and the evaluation results show clear performance benefits.
Therefore, I would recommend an accept for this paper. I do have a few questions though and it would be great if the authors could address them.

In terms of the global scheduler for scheduling requests, I am a bit confused about why Preble chooses to favor exploitation over exploration on the point when the amount of recomputation saved (i.e. number of tokens in the matched prefix) is larger than the amount of new computation (i.e. number of tokens in the remaining prompt).
As far as I am concerned, if a request has a partial overlapping prompt with the prefix cache, it seems to always be beneficial to use the prefix.
This is because, as pointed out in the paper, the prefill latency grows roughly linearly with the number of tokens, so saving the number of tokens needing to compute could reduce prefill latency.
It is true that doing so naively may lead to load imbalance between GPUs, but my feeling is that the load within different GPUs holding the same prefix should play a part in determining whether the scheduler should explore or exploit, instead of merely using the number of tokens as the determining condition.

I am also not sure if the workloads chosen in the paper could fully reflect the real-world distributed LLM serving setting.
How would the sharing ratio be on some chat datasets such as ShareGPT or LMSYS?
In addition, the current evaluation is done on the five datasets respectively.
However, in a real serving scenario, there may be a set of LLM applications running on top of the same LLM backend at the same time.
It may be beneficial to show how Preble performs when requests are drawn from different datasets.

Other questions:
1. In Sec 3.2, it says "If an existing tree node only matches partially to the new request, we split the node into the matched part and the remaining part." How often does this happen and would this split incur much overhead?
2. For the case of "exploitation", the algorithm first selects GPUs that have the tree node with the longest tokens in the matched prefix, then selects the GPU with the lightest request load among them. Is this always optimal? What about the case that you have a GPU that has the second longest tokens in the matched prefix but an even lighter load?
3. I find it a bit unintuitive that the value of time window H does not affect the end-to-end results. Have any evaluations been done to evaluate the effect of different time window values?
4. In Sec 4.4, it says "Other workloads and distributions benefit from a different set of techniques." What do you mean by that?
5. In Sec 4.4, it says "One Preble global scheduler can sustain at least 70 to 391 concurrent A100 GPUs", Could you kindly explain how these numbers are calculated and obtained?

---

> ### Author Response · Authors · 2024-11-20
>
> Thank you for your insightful question regarding the global scheduler's balance between exploitation and exploration in Preble. Below, we address your concerns and clarify the reasoning behind our approach.
>
> **Policy for Deciding Exploitation vs. Exploration**
>
> Thank you for your insightful discussion. We want to clarify that with our E2 algorithm (Section 3.2, Algorithms 1 and 2 on pages 19 and 20), because of our way of calculating per-GPU load cost, requests categorized as exploration can still reuse shared prompt prefixes, and requests categorized as exploitation still consider load. For the requests in the exploration category, the last load-cost term considers the overhead of recomputation for running the request on a GPU with no cached prompt. Thus, when the other two terms do not dominate (when the cluster is overall balanced or when the cluster has low load overall), the request will still be directed to a GPU with a cached prompt and exploit the cached value. On the other hand, for requests under the exploitation category, we do consider the load within different GPUs holding the same prefix and choose the lightly loaded one, as you suggested. Our E2 scheduling policy balances the load factor and the prefix-reusing factor in a unified way, which effectively improves the overall performance of the system, as shown in our experimental results. That said, we agree that exploring different criteria for exploitation vs. exploration is an interesting research topic. Please check out our answer to Reviewer 3qUC for more discussion around this topic.
>
> **Question: Real World Workloads and Mixed Workloads**
>
> Thank you for the suggestion. We already evaluated a mix of workloads using a real-world trace arrival pattern. In Figure 4, we evaluate Preble’s handling of mixed workloads (Tool + Video) using the BurstGPT arrival pattern. ShareGPT and LMSys datasets often contain shorter conversations with fewer shared prefixes, as they don’t employ augmented/complex prompting. As a result, Preble may not enable as much benefit as our evaluated workloads. In the worst case, Preble degenerates to the same scheduling policy and thus performs similarly as SOTA distributed systems if there is no prompt sharing at all. However, popular platforms such as Anthropic [1], OpenAI [2], and Gemini [3] demonstrate the significant economic and efficiency advantages of prefix sharing. Thus, we expect Preble to be useful in most production systems.
>
> **Q1: Radix tree split overhead**
> The cost of splitting a tree node is minimal, requiring a single index select operation and an object creation. The most popular prefixes will be split first. While not the focus of the work, this can be further optimized with faster mechanisms. The frequency of a split depends on the workloads. For workloads that have a lot of branching patterns(system prompt, few shot examples, tool descriptions) such as Toolbench, splits might occur more often. For workloads that have minimal branching such as LooGLE, splits are more rare.
>
> **Q2: Exploitation load calculation**
> For the exploitation policy, we currently use a greedy policy that selects the GPU with the largest prefix match. If there are multiple GPUs all sharing the same largest prefix, we choose the one with the lightest load. This greedy policy works well for our tested workloads and load patterns. Admittedly, there may be other cases where the greedy policy does not work as well as a globally optimal solution. For such cases, Preble’s current solution is to perform its rebalancing, which would reduce the load imbalance caused by the greedy exploitation approach. We leave the exploration of other exploitation policies, such as what you suggested, to future work.
>
> **Q3: Effect of Time Window H**
> Preble uses the time window to predict the future load of a GPU, which is part of the load cost calculation for each GPU. As we only use the relative load-cost ranking across GPUs to determine the selection of GPU, the absolute load (and thus how well we predict) is less critical, as long as we use the same window size across GPUs. To validate this, we provide some sample results of varying the time window below using the Toolbench workload with the BurstGPT arrival pattern over 10 mins:
>
> | H Time Window | Average Request Latency (sec) |
> |---------------|--------------------------|
> | 5 min | 9.72 |
> | 4 min | 9.53 |
>  | 3 min | 9.94 |
>  | 2 min | 10.11 |
>  | 1 min | 9.14 |
> | 10 sec | 7.41 |
>
> For reference, the average request latency for Round Robin is 37 seconds. Compared to the Round Robin policy, our results are relatively insensitive to the value of H. Future works can explore dynamic settings of the window size, e.g., for more bursty or more frequently changing arrival patterns.
>
> **References:**
>
> [1] https://openai.com/index/api-prompt-caching/
>
> [2] https://docs.anthropic.com/en/docs/build-with-claude/prompt-caching
>
> [3] https://ai.google.dev/gemini-api/docs/caching

---

> > ### Author Response · Authors · 2024-11-20
> >
> > **Q4: Clarification on Ablation**
> >
> > The ablation study in Section 4.4 used the Toolbench workload with a Zipf distribution of prefix requests at an RPS of 36. Inter-request arrival times sampled from a Poisson distribution introduce temporal burstiness, leading to request clustering that can overload specific nodes. Post-assignment request migration effectively mitigates this by redistributing the load. Prefill-decode balancing is particularly impactful when prompt/decode ratios vary significantly, as seen in the programming workload. Local waiting queue ordering is crucial for skewed prefix-sharing distributions, balancing fairness and cache reuse to prevent starvation of low-cache-hit requests, and improving p99 latency without affecting average latency.
> >
> > **Q5: How is the scalability calculated**
> >
> > As we do not have the resources to scale to hundreds or thousands of GPUs, we estimate our global scheduler’s scalability by sending an extremely high load of requests to the global scheduler. The global scheduler performs all its scheduling tasks and manages a global radix tree that would be constructed with *k* GPUs. We measure the maximum number of requests the global scheduler can schedule without violating the SLO (max request latency allowed), and we divide this number by the single-GPU request load to obtain the number of GPUs the global scheduler can sustain. We do not perform the actual GPU computation, as we do not have hundreds of GPUs and because each GPU in a large cluster would still perform the same amount of work as in a small cluster. As discussed in our response to Reviewer **3qUC**, our scalability test focuses on the global scheduler as the main scalability bottleneck. Other factors, such as networking jitters and server failure, can still happen on a large scale, but their measurement is out of the scope of this paper.
> >
> > Thank you again for your thoughtful review and questions. We greatly value your insights and look forward to any further feedback that can help refine and expand this work.

---

> > > ### Comment · Reviewer_NYyJ · 2024-11-28
> > >
> > > Thank you for your response. The detailed clarification and experiment results on window size address my questions. I would like to keep my score.

---

### Official Review · Reviewer_3qUC · 2024-11-02

**Soundness:** 4
**Presentation:** 3
**Contribution:** 3
**Rating:** 8
**Confidence:** 4

**Summary:**

This paper introduces a distributed LLM inference framework, Preble, which is specifically designed for long and sharing prompts. Before diving into the solution, the paper first characterizes a wide range of LLM applications and production LLM inference traces that motivate the problems. By designing E2, a distributed LLM request scheduling algorithm, the paper aims to improve LLM usage with long and sharing prompts. Extensive experiments are performed against two popular LLM inference systems: SGLang and vLLM. By running real-world applications, workloads, and GPU testbeds, Preble outperforms existing baselines by significantly improving LLM serving performance.

**Strengths:**

1 - The paper is well-written and well-organized. I appreciate the intuitive diagrams and figures in the paper.

2 - Motivations and characterizations are clear and inspiring.

3 - The design and analysis are technically clear and sound.

4 - The proposed framework is implemented and evaluated against two popular LLM inference systems, vLLLM and SGLang.

5 - Extensive experiments are conducted, and real-world LLM inference applications are used in the evaluation, which is comprehensive.

**Weaknesses:**

1 - E2 determines whether it's exploration or exploitation by measuring the computation for matched tokens and computation for unique tokens. This simple design is intuitive enough but remains fixed across different cases. It might be better to make this design dynamic with a knob with load-awareness, where we measure the ratio alpha = (computation for matched tokens / computation for unique tokens) as the knob. For example, if the overall GPU load is quite low (e.g., request rate is low), it would make sense to favor exploitation more by setting this knob alpha lower than 1. This retreats to your fixed design when alpha = 1.

2 - " E2 calculates all three costs as GPU computation time and finds the GPU with the lowest sum." Would it be better if this could be designed as a weighted sum to prevent any dominant components out of the three parts?

3 - The paper mentions that it provides prefill-decoding balancing. How does Preble interact with or integrate with existing continuous batching [1] and chunked prefill [2, 3] techniques?

4 - Minor writing issues: in Appendix C, Figure 11, SGLang should be vLLM in the legends?

[1] Kwon, Woosuk, et al. "Efficient memory management for large language model serving with pagedattention." Proceedings of the 29th Symposium on Operating Systems Principles. 2023.

[2] Zhong, Yinmin, et al. "{DistServe}: Disaggregating Prefill and Decoding for Goodput-optimized Large Language Model Serving." 18th USENIX Symposium on Operating Systems Design and Implementation (OSDI 24). 2024.

[3] Agrawal, Amey, et al. "Taming {Throughput-Latency} Tradeoff in {LLM} Inference with {Sarathi-Serve}." 18th USENIX Symposium on Operating Systems Design and Implementation (OSDI 24). 2024.

**Questions:**

Please see questions in weaknesses.

---

> ### Author Response · Authors · 2024-11-20
>
> **Dynamic Adjustment of Exploration vs. Exploitation with Load-Aware Knob**
>
> Thank you for your insightful observation and suggestion.
>
> A subtle point in our design that may not have come through in our paper writing is that even if a request is categorized as exploration, it may still be sent to a GPU with its shared prefix. This is especially true when the overall cluster GPU load is low. This is because our load cost calculation has three terms: the overall request load, the eviction cost, and the cost to run the current request. When the cluster load is low, the first two terms will also be low, making the current request’s computation cost the dominating factor. In this case, the GPU with more shared prefix will have a low computation cost for the current request, and E2 will choose that GPU.
>
> To see how a load-aware knob could potentially work, we ran an experiment using the maximum cluster utilization as a knob, as you suggested. We found the results similar or slightly worse than the original Preble (see https://imgur.com/a/SEXvX5B). For example, in the LooGLE dataset, the default Preble performs (min: -4%, max: 9.2%, avg: 3%) better than the knob-based Preble. In the Toolbench dataset, the default Preble performs  (min: -1.6%, max: 15%, avg: 5.43%) than the knob-based Preble.
>
> As a future direction, it is totally valid to explore load-aware policies that adjust the exploration-exploitation balance, especially for handling unknown burst patterns where aggressive exploitation could be detrimental.
>
> **Using weighted sum in the cost equation**
>
> The load calculation measures all types of overhead in a unified unit: the GPU computation load. When a GPU has high utilization, the first factor (overall load) dominates the other two factors (eviction cost and new-request computation), which is exactly what we want, as the request assignment should mainly consider the overall GPU loads. When a GPU has extremely low utilization, the new-request computation factor dominates, which is then again what we want, as the assignment of GPU should mainly consider the new request and possibly favor a GPU with more shared prefix instead of considering overall GPU load. This unified measurement is versatile in various scenarios, and our experiments have shown it to work empirically well. That said, considering weighted sums for the cost calculation is possible and something future works could explore more.
>
> **Integration with Continuous Batching and Chunked Prefill Techniques**
>
> These techniques are already used internally in Preble. Preble uses continuous batching in its local scheduler (Section 3.3). Preble also performs chunked prefill at its local scheduler. On top of these local-level techniques, we propose the prefill-decoding balancing at our global scheduler by balancing GPUs heavily loaded with decoding computation with new requests that have been determined to explore (i.e., will go through prefill recomputation). We do not perform prefill-decoding disaggregation like what’s used in DistServe, because requests with different prompt sharing degrees have different prefill/decode ratios and benefit more from our global prefill-decoding balancing plus local chunked prefill.
>
> **Minor Writing Issue in Appendix C**
>
> We appreciate your careful review. You are right, and we corrected the labeling of that plot in the current revision.

---

> > ### Comment · Reviewer_3qUC · 2024-11-28
> >
> > Thank you for the answers. They have addressed my questions.

---

### Official Review · Reviewer_tC1t · 2024-11-04

**Soundness:** 3
**Presentation:** 3
**Contribution:** 3
**Rating:** 6
**Confidence:** 3

**Summary:**

The paper presents a distributed serving system designed to handle large language models (LLMs) with long and shared prompts efficiently. Preble aims to improve the reuse of computed key-value (KV) state across multiple GPUs, overcoming the limitations of current systems that primarily focus on single-GPU optimization. The authors introduce a new distributed scheduling algorithm called E2 (Exploitation + Exploration) to balance the reuse of shared prompt prefixes while managing GPU load effectively. Evaluations using real-world workloads demonstrated that Preble outperforms existing state-of-the-art systems by reducing average latency and p99 latency.

**Strengths:**

- The authors propsed a first system to address efficient prompt-sharing in distributed LLM environments.
- The proposed E2 scheduling algorithm seems to effectively balance prefix cache sharing and computation load across GPUs and the authors show significant performance gain compared to SGLang.

**Weaknesses:**

- Althought the paper claim strong scalability, it is only partially supported by experiments on two four-GPU machines.

**Questions:**

- What are potential unobserved/unexpected bottlenecks when the systems scales to hundreds or even thousands of machines? Wiil they significantly limit the scalability of the system?

- Given that prompt-sharing can vary beyond exact prefixes, can Preble be extended to support non-prefix caching mechanisms?

- How does Preble’s fairness mechanism prioritize requests if there are multiple users with varying priorities?

- Is there any Trade-off between Cache Utilization and Load-Balancing in E2 Scheduling?

---

> ### Author Response · Authors · 2024-11-20
>
> Thank you for your insightful questions. We address them below.
>
> **Experiments Beyond four GPUs**
>
>
> Because of budget and resource constraints, we were unable to perform large-scale experiments at the time of the submission. Below, we provide experimental results on 2 to 16 A6000 GPUs running the LooGLE dataset. As we increase the number of GPUs in the cluster, we also increase the number of unique prefixes proportionally, so the prefix tree (and thus the amount of work that Preble’s global scheduler needs to handle) also grows proportionally. We report the maximum sustainable request per second (rps) when meeting an average request latency target of 20 seconds. As seen, even though the global prefix tree grows its size linearly, Preble still achieves linear scalability, while the simple round-robin scheduler is constantly slower and scales much worse than Preble. You could also view the detailed result figures at https://imgur.com/a/puhNFdX and in the revised Appendix.
> | Policy       | GPUs = 2 | GPUs = 4 | GPUs = 8 | GPUs = 16 |
> |--------------|----------|----------|----------|-----------|
> | ROUND_ROBIN | 0.4      | 0.8      | 1.5      | 3.0       |
> | Preble       | 0.7      | 1.4      | 2.79     | 5.7       | 11.1 |
>
> **Unexpected bottlenecks to scalability**
>
>
> As discussed in our paper, from our current experiments and measurement, maintaining and accessing the global prefix tree at the global scheduler is the main scalability bottleneck, and our implementation techniques like synchronization-free data structure have already largely addressed this issue. Future improvements like using a native programming language instead of Python could further improve the global scheduler’s scalability.
> Apart from that, there could be several factors that affect the scalability of Preble and other distributed serving systems. For example, at the data-center scale, network congestions and server failures could happen more often than in our test environment. Heterogeneity (e.g., different GPUs, different topology) is another challenge for scalability. These factors have been studied in traditional distributed computing systems, and we leave the exploration of them for future work.
>
> **Can Preble be extended to support non-exact prefix matching?**
> While not the focus of our current work, extending Preble to support non-exact prefix matching like [1,2] is totally possible and could result in larger benefits. Our E2 scheduling algorithm and Preble’s scheduler architecture will still apply to non-exact prefix matching. However, the prefix tree matching operation needs to be replaced with the alternative non-exact matching operations.
>
> **Fairness Mechanism for Multiple User Priorities:**
>
> Currently, Preble does not support explicit priority handling for users with different priority levels. A potential extension to Preble is to assign a priority level to each request and use it as an indicator for both the global and local schedulers. The global scheduler can enforce a weighted allocation strategy, where each priority level is given a proportionate quota of resources (e.g., a percentage of available GPUs) based on the system load and the user demand. This would ensure that higher-priority requests experience lower latency and more resources to cache prefixes. At the local scheduler, each request can be placed in a certain group based on its assigned priority level, with the scheduling of intra-group requests being the same as the current Preble.
>
> **Trade-off Between Cache Utilization and Load-Balancing in E2 Scheduling:**
>
> As explained in Section 3.2, high cache utilization could lead to skewed GPU loads, implying high request tail latencies and low overall cluster GPU utilization. Meanwhile, an evenly balanced load could imply more recomputation for not reusing shared prefix cache, resulting in delayed user request latencies. Preble gets the better out of the two approaches with our E2 algorithm.
>
>
> **References:**
>
> [1] In Gim, Guojun Chen, Seung seob Lee, Nikhil Sarda, Anurag Khandelwal, and Lin Zhong. Prompt cache: Modular attention reuse for low-latency inference. In Proceedings of the 7th MLSys Conference, Santa Clara, CA, May 2024.
>
> [2] Liu, Shu, Asim Biswal, Audrey Cheng, Xiangxi Mo, Shiyi Cao, Joseph E. Gonzalez, Ion Stoica and Matei A. Zaharia. Optimizing LLM Queries in Relational Workloads. ArXiv 2024

---

### Official Review · Reviewer_ck7x · 2024-11-05

**Soundness:** 3
**Presentation:** 2
**Contribution:** 2
**Rating:** 6
**Confidence:** 4

**Summary:**

The paper presents Preble which optimize for prompt sharing in distributed LLM serving system. More specifically, the paper proposed some load-balancing mechanism that takes considerations into prompt sharing and fairness for large language model in multi-GPU setting, and designed global and local scheduler for coarse assignment and fine-grained scheduling and eviction policy. Experiments show the performance gain comparing to vLLM and SG-lang based baseline.

**Strengths:**

The paper achieved performance improvements comparing to prompt caching mechanisms (vLLM and SG-lang).

The paper further considered fairness into scheduling process in the optimization in prompt cache.

**Weaknesses:**

The major concern is that contribution and novelty of this paper is limited.

Distributed prompt sharing optimization across GPUs: Optimize the inference among multiple GPUs (even in cluster level system) is previously studied by various work. As some of the previous work has been cited by the paper, mem-serve[1], inference without interference [2], mooncake[3] are all large scale systems serves beyond a single GPU. Those systems all considered shared prompts, and can be used with long context in production level. In those papers, global scheduling algorithms are already proposed to route the request with minimal re-computation. Yet, the method is only compared with vLLM and SG-Lang. The paper didn’t discuss the difference between those serving system, as well as considering them as baselines to discuss.

The localized fine-grained local scheduler: The design of the local scheduler is mainly based on Radix-tree scheduling (SG-Lang), considering fairness added on.

The scenario of the prompt sharing: Previous literature SG-Lang[4], CacheBlend[5] and Prompt Cache[6] have all mentioned the cache reuse opportunity.

[1] MemServe: Context Caching for Disaggregated LLM Serving with Elastic Memory Pool
[2] Inference without Interference: Disaggregate LLM Inference for Mixed Downstream Workloads
[3] Mooncake: A KVCache-centric Disaggregated Architecture for LLM Serving
[4] SGLang: Efficient Execution of Structured Language Model Programs
[5] CacheBlend: Fast Large Language Model Serving for RAG with Cached Knowledge Fusion
[6] Prompt Cache: Modular Attention Reuse for Low-Latency Inference

**Questions:**

See the weakness.

---

> ### Author Response · Authors · 2024-11-20
>
> Thank you for referring us to these related works. Our arXiv and GitHub releases (May 8) **predate** MemServe, Mooncake, and CacheBlend. Among the related works mentioned, *Inference without Interference* does not consider prompt sharing in its global scheduling; *MemServe* considers prompt sharing but not together with load distribution. Thus, both perform worse than Preble. *Mooncake* is a close-sourced production system that was heavily influenced by Preble. *CacheBlend* and *Prompt Cache* perform non-exact prompt caching and do not work at the distributed scheduler level.
>
> Overall, while various previous and concurrent works have adopted prefix caching, they do not consider intricate factors like long-term load impacts in a distributed system. Being the **first prompt-cache-aware distributed LLM serving system**, Preble’s key contribution lies in its E2 scheduling algorithm that balances prompt sharing and load distribution, which is the key to Preble’s superior performance over these other works. Below, we provide a more detailed comparison of Preble to MemServe and Mooncake, as they are the two most related pieces of work in the list.
>
> **Comparison to MemServe**
>
> MemServe is a system that focuses on the disaggregation of prefill and decoding steps. Its global scheduler uses a global prefix tree and matches each request to the longest common prefix in the tree. If a match is found, MemServe always sends the request to the first GPU that caches the matched prefix, even when the prefix is short (e.g., a few system-prompt tokens). This scheme results in a high skew of loads across GPUs and degraded performance. In contrast, Preble considers both load and prefix sharing in its E2 scheduling algorithm, and we employ other techniques like load rebalancing and prefill-decoding balancing to further improve the overall system performance.
>
> We compare Preble with MemServe using the LooGLE dataset on two and four GPUs. As seen below (and added to the Appendix C Figure 13 of the revised paper), MemServe performs significantly worse than Preble and under various workloads.
>
> **LooGLE 2 GPUS Average latency**
> | Scheduler       | rps = 0.3 | rps = 0.5 | rps = 0.7 |
> |--------------|-----------|-----------|-----------|
> | MemServe    | 48.09     | 96.5      | 117.8     |
> | Preble       | 7.99      | 15.57     | 18.85     |
>
> **LooGLE 4 GPUS Average Latency**
> | Scheduler       | rps = 0.3 | rps = 0.5 | rps = 0.7 |
> |--------------|-----------|-----------|-----------|
> | MemServe    | 185.9           | 247.68 | 295.75 |
> | Preble       | 7.43            | 12.58 | 16.27 |
>
> **Toolbench 2 GPUS Average latency**
> | Scheduler       | rps = 6 | rps = 9 | rps = 12 |
> |--------------|-----------|-----------|-----------|
> | MemServe    |  103.16 | 224.6 | 396.1 |
> | Preble       | 3.24 | 7.24 | 18.68 |
>
> **Toolbench 4 GPUS Average latency**
> | Scheduler       | rps = 12 | rps = 18 | rps = 24 |
> |--------------|-----------|-----------|-----------|
> | MemServe    |  395.7  | 744.85 | 1222.3
> | Preble       | 3.71 | 11.34 | 27.15
>
>
> **Discussion on Mooncake**
>
> Mookcake is a production system (powering the chat product https://kimi.moonshot.cn/) that is influenced by and released after our work. Their prefix- and load-based scheduling approach follows Preble’s scheduling approach. They cited our work in Section 8 by acknowledging their similarities in addressing prefix caching challenges. A key limitation of Moonshot is the fixed threshold they use to determine whether or not to exploit prefix sharing, as one threshold cannot work well across different requests. In contrast, Preble’s E2 algorithm determines exploitation vs. exploration based on each request’s own sharing feature. Thus, Preble’s solution should work better across requests and workloads. However, as a production system, Moonshot is close-sourced and does not release the hardcoded threshold they use. As such, we are unable to directly compare Preble with Moonshot.

---

> > ### Author Response · Authors · 2024-11-22
> >
> > We hope this message finds you well. As we approach the final week of the discussion phase, which concludes on November 26th AoE, we kindly request your feedback on our rebuttal. If you find our rebuttal helpful or informative, we would greatly appreciate your recognition by increasing your review rating. Please don’t hesitate to reach out with any questions or if further clarification is needed.

---

> > > ### Comment · Reviewer_ck7x · 2024-11-23
> > > **Thanks for the rebuttal**
> > >
> > > Thanks to the author for the rebuttal especially on the results comparing with men-serve. In light of the justification of the novelty regarding distributed prompt sharing optimization across GPUs. I raised my point.
> > >
> > > For the implementation details, there are some points that I would like to discuss and understand more. If I understand it correctly, preble builds a scheduler to dispatch the request. Given that the underlying vllm and sglang(backended with flashinfer for prefix sharing) already use prefix cache in computing, preble doesn’t need to modify any underlying CUDA kernel, but just add a router to route the request to the GPU, and prefix sharing is then handled by flashinfer(as with Sglang). Sort of like given earlier request is computed, the kv cache prefix should be cached, and if the newer request with the same prefix comes right after the previous requests before their caches are evicted, then the cache is reused. In such case, it is possible that the underlying cache has been evicted. This is similar to Sglang that add a scheduler for high level request scheduling and doesn’t need to modify the underlying flashinfer backbone  for computing in the radix tree cache. Is that correct?

---

> ### Comment · Reviewer_ck7x · 2024-11-24
> **Further question for understanding contribution**
>
> Here I have a further high level question to discuss, I think I can interpret the technical contribution as a scheduler considering SgLang + load balance between the GPU's in the E2 algorithms? Basically, SGlang did the prefix cache scheduling through lower level FlashInfer for prefix cache, and Exploitation inherent from there and load balancing the request to other GPUs with low utilizations (Exploration). Thanks

---

> > ### Author Response · Authors · 2024-11-24
> >
> > Thank you for your thoughtful comments and for raising the score. Your understanding is primarily correct.
> >
> > Preble is a global scheduler that routes requests to different GPUs based on the computation load **and** cached prefix states on each GPU. The key contribution of Preble is its E2 scheduling algorithm, which considers prefix matching (like SGLang) and load balance, as you interpreted. Preble has several other technical contributions such as its rebalancing policy, prefix-caching-based prefill-decode balancing, local-scheduler fairness, etc.
> >
> > As you mentioned, cached prefixes can be evicted before a subsequent request runs. The global scheduler will be notified about any eviction events on the local scheduler, as detailed in Sec 3.3, to update the prefix tree for scheduling future requests.

---

### Official Review · Reviewer_rygC · 2024-11-08

**Soundness:** 1
**Presentation:** 3
**Contribution:** 2
**Rating:** 3
**Confidence:** 3

**Summary:**

The paper introduces Preble, a distributed LLM serving system specifically designed to handle long-context prompt workloads. The authors propose a tree-based prefix-matching algorithm and an online workload-balancing algorithm that combines exploration and exploitation strategies. Preble demonstrates the ability to reduce P99 latency for the Mistral 7B and LLaMA-3-70B models on video QA workloads, using Azure’s LLM request arrival patterns, compared to SGLang and vLLM.

**Strengths:**

The scheduling algorithm seems interesting. The authors introduce a lot of background of LLM in the appendix.

**Weaknesses:**

I think the major problem is that the authors mix the concept of high-level cluster scheduling with low-level GPU kernel design, which is very confusing. In Section 3.1, the overall design, the term “data parallel” is used inaccurately. Typically, data parallelism refers to the process of dividing data into chunks and distributing these chunks across blocks and threads in low-level kernel design, instead of splitting data to different servers at a high level.

The statement in the abstract, “production LLM serving systems are distributed by nature,” is misleading. Whether a production LLM requires distributed serving depends on the model size. For instance, in the evaluation section, it’s unclear why the Mistral 7B model is split across multiple servers, as it can easily fit and run inference on a single A100 GPU.

The authors claim to have implemented their scheduling system on top of vLLM, but there is no explanation of this implementation, nor does vLLM appear in any of the evaluation results. To my knowledge, vLLM supports distributed serving and leverages Ray for efficient workload balancing.

**Questions:**

In Section 3.1, it states, “Our current implementation of Preble scales to at least 70 to 391 GPUs.” Based on my understanding, Preble is designed as a single model serving system. Why, then, does it require such a large number of GPUs to serve a single model? I can serve a LLaMA-3-70B model using just four A100 80GB GPUs.
How does Preble compare to vLLM?

---

> ### Author Response · Authors · 2024-11-20
>
> We understand your confusion about the term "data parallelism." Below, we explain what we mean by “data parallelism,” its production usage, and our vLLM results.
>
>
> **Meaning of Data Parallelism**
>
> In the context of model serving, data parallelism usually refers to **launching multiple replicas of a model on different GPUs** to serve different **(data) requests** in parallel for a higher overall system throughput (higher number of requests completed per second). This definition has been used extensively in research papers [1,2,3], in vllm[7] and industry practices [8, 9]. We follow the same nomenclature when talking about data parallelism throughout the paper.
>
> Differently, kernel-level parallelism, as mentioned in your review, is a technique used within a single GPU that improves the performance of a single request, and tensor parallelism allows for model weights to exceed the capacity of a single GPU. Both are orthogonal and can be used together with data parallelism. For example, if a model needs two GPUs to run, and it is desired to run three sets of data in parallel, we would have (GPU1, GPU2 => first model replica), (GPU3, GPU4 => second replica), (GPU5, GPU6 => third replica).
>
> **Distributed LLM Serving with Data Parallelism in Production**
> Our work focuses on building an efficient distributed LLM serving system that employs data parallelism. Data parallelism is common in production inference systems. Providers like OpenAI and Anthropic deploy hundreds or thousands of replicas of models across clusters of GPUs to meet high user request load, even when a model (e.g., "Mistral 7B") fits a single GPU. This is what we mean by “production LLM serving systems are distributed by nature” and why being able to support a large number of GPUs is important.
>
> **Implementation and results of vLLM**
> Because of the space constraint, we were unable to include vLLM results in the main text, but they are included in the Appendix on Page 21 (Figure 11 and Section C). We had a typo in Figure 11’s label; the result orange lines in that figure are for vLLM instead of SGLang. We apologize for that and have fixed the labeling issue in the current revision.
> We implemented our scheduler on top of vLLM by managing different replicas of vLLM with data parallelism. No Ray or other distributed schedulers were used. The Preble global scheduler directs requests to different GPUs running vLLM. Overall, we achieve up to 2x lower request latency than vLLM.
>
> **References:**
>
> [1] Liu, Yiheng, et al. "Understanding LLMs: A comprehensive overview from training to inference." arXiv preprint arXiv:2401.02038 (2024).
>
> [2] Jia, Zhihao, Matei Zaharia, and Alex Aiken. "Beyond data and model parallelism for deep neural networks." Proceedings of Machine Learning and Systems 1 (2019): 1-13.
>
> [3] Xiang, Yecheng, and Hyoseung Kim. "Pipelined data-parallel CPU/GPU scheduling for multi-DNN real-time inference." 2019 IEEE Real-Time Systems Symposium (RTSS). IEEE, 2019.
>
> [4] Sun, Zekai, et al. "Hybrid-Parallel: Achieving High Performance and Energy Efficient Distributed Inference on Robots." arXiv preprint arXiv:2405.19257 (2024).
>
> [5] https://lmsys.org/blog/2024-07-25-sglang-llama3/
>
> [6] https://mlsys.wuklab.io/posts/scheduling_overhead/
>
> [7] https://github.com/vllm-project/vllm/issues/9206
>
> [8] https://developer.nvidia.com/blog/demystifying-ai-inference-deployments-for-trillion-parameter-large-language-models/
>
> [9] https://www.infracloud.io/blogs/inference-parallelism/

---

> > ### Comment · Reviewer_rygC · 2024-11-20
> > **Confusion about vLLM implementation**
> >
> > Thank you for your detailed response.
> >
> > However, my confusion goes beyond the semantic interpretation of “data parallel.” I am unclear about how PREBLE was actually implemented on top of vLLM. Specifically, if PREBLE involves prefix sharing, wouldn’t this require kernel modifications to properly manage the KV cache? Could you clarify the implementation details and any adjustments made to vLLM’s architecture to support these features?
> >
> > Additionally, the vLLM results, despite being a critical benchmark, were placed in the appendix and mislabeled as SGLang. Given their importance, they should be included in the main experimental results for a more transparent and meaningful comparison.
> >
> > Thank you again for addressing these concerns.

---

> > > ### Comment · Reviewer_rygC · 2024-11-20
> > > **Further Clarification**
> > >
> > > There is a relevant discussion in the vLLM GitHub repository:
> > > https://github.com/vllm-project/vllm/issues/227
> > >
> > > In this issue, the maintainers mention:
> > > “Thanks for bringing this up! Indeed, prefix sharing is an excellent scenario to save even more memory and compute. We evaluated this setting in our research paper. However, our current implementation of the PagedAttention kernel with query sequence length > 1 is buggy and slow, so we didn’t include it in our original release. We plan to add this feature in the future.”
> > >
> > > Thank you again for your time and effort in addressing these points.

---

> > > > ### Author Response · Authors · 2024-11-20
> > > >
> > > > Thank you for your quick reply and your future questions. We clarify them as follows.
> > > >
> > > > Preble is a distributed system that sits on top of inference systems like vLLM and SGLang. These inference systems already support prefix caching and sharing on a per-GPU-node base. For example, vLLM added the support for prefix caching in v0.3.4 [1, 2]. vLLM retrieves cached KV state for matching prefixes and passes the KVs to its attention kernel as pre-computed context. We do not change this part or any other kernel-level implementation. Preble’s global scheduler implementation is completely isolated from inference systems like vLLM and SGLang. We added the network communication support to vLLM so that each node can talk to the global scheduler (e.g., to accept requests sent by the global scheduler, to send the request eviction information to the global scheduler. Other than that, we modified vLLM’s wait queue implementation to add fairness consideration. All these changes are peripheral to vLLM, and we do not touch its core.
> > > >
> > > > As for the GitHub issue you brought up, vLLM initially had a performance regression when first introducing prefix caching [2]. Its performance is greatly improved after vLLM switched to the FlashInfer kernel [3]. As we do not change vLLM core, and our focus is on data-parallel distributed system performance, we simply compare our extension of vLLM to vLLM extended with Round Robin scheduling using the same vLLM version (v0.3.4).
> > > >
> > > > We presented our performance comparison with SGLang in the main text, because SGLang performed better than vLLM at the time of writing [4]. Additionally, vLLM followed SGLang’s prefix caching idea after SGLang, and prefix caching is SGLang’s main technical contribution.
> > > >
> > > >
> > > > **References:**
> > > >
> > > > [1] https://docs.vllm.ai/en/latest/automatic_prefix_caching/apc.html
> > > >
> > > > [2] https://github.com/vllm-project/vllm/issues/2614
> > > >
> > > > [3] https://flashinfer.ai/2024/02/02/cascade-inference.html
> > > >
> > > > [4] https://lmsys.org/blog/2024-07-25-sglang-llama3/

---

> > > > > ### Comment · Reviewer_rygC · 2024-11-21
> > > > > **Clarification on implementation**
> > > > >
> > > > > Yes, this is a descent explanation for the implementation. Thanks. Prefix sharing is very useful for saving memory on a single GPU by enabling multiple threads to access the shared KV cache, with kernel design often being the primary challenge. In your context, it seems that “prefix sharing” refers to a preprocessing step that matches requests based on their content and schedules as many requests with the same prefix as possible to the same GPU server. You are calling APIs of inference engines such as vLLM, SGLang, without key modifications to them. It would be helpful to make this distinction clear in the paper, instead of claiming "implementing on top of vLLM".
> > > > >
> > > > > Additonally, the "prefix" in single GPU prefix sharing is the key, query, value after encoding and operator operations. In this case, are the server providers allowed to inspect the content of user requests without violating privacy policies? If the “prefix” you are matching corresponds to the sentence content, does this effectively align with the processed keys, queries, and values stored in the KV cache?

---

> > > > > > ### Author Response · Authors · 2024-11-21
> > > > > >
> > > > > > Thank you for taking the time to read and understand our responses. Your high-level understanding of our work is absolutely right, and we will clarify Preble’s relationship to vLLM in our future version of the paper. There is just one small clarification regarding how we schedule requests with the same prefix: Preble does not always send a request with a shared prefix to the GPU that caches the context; instead, it weighs in the overall cluster GPU utilization and prioritizes load-balancing if needed even when that means not sending the request to a cached GPU. These details are discussed in Section 3.2 as part of our E2 scheduling policy.
> > > > > >
> > > > > > As for your new questions, we want to first clarify that what we (and other single-GPU works like SGLang and vLLM) are matching is the tokenized user requests. If the user request has prefix tokens that match a previously seen and cached request, we (and others) call that a prefix matching. For example, one request is “You are a helpful assistant in answering questions. My question is what year is this year”, and another request is “You are a helpful assistant in answering questions. My question is who won the US election in 2024”. Prefix matching will find that the first 11 words are the same. When the system processes the first request and computes K and V for it, the KV are cached. When the second request comes, if we were to compute it from scratch, the first 11 words would result in the exact same KVs as the first request; so the system just reuses the previous KV.
> > > > > >
> > > > > > In terms of privacy, providers have to know the exact request content to perform their model generation. Preble does not inspect any additional information. In production systems like in OpenAI [1], prefix caching is enabled by default for all consumer users.
> > > > > >
> > > > > > We kindly request that, after addressing all your concerns, the score be revised to reflect your responses to our rebuttal.
> > > > > >
> > > > > >
> > > > > > [1] https://platform.openai.com/docs/guides/prompt-caching

---

> > > > > > > ### Comment · Reviewer_rygC · 2024-11-24
> > > > > > > **Clarification on implementation**
> > > > > > >
> > > > > > > But before I can raise the score, I want to have further discussion as I have a strange feeling both at a first glance and after re-reading the paper. Although the justifications all seem reasonable, I am still not quite sure about how actually the system works, maybe partly because I am not very familiar with SGLang.
> > > > > > >
> > > > > > > Could you address reviewer ck7x's further question? The new baseline you compare, MemServe [1], is implemented on top of vLLM: "To realize this design, we make minor changes to an existing inference engine (e.g., vLLM). As a result, the prefill instance will call MemPool’s transfer API to transfer the active KV cache produced after the prefill phase to the decode instance. We carry essential metadata the decode instance requires in transfer’s private field, such as request ID, sampling parameters, prompt tokens, etc."
> > > > > > >
> > > > > > > "We adapt vLLM to using MemPool APIs. Specifically, we replace its original cache engine and hash-based prefix caching with MemPool. To realize block aggregation, we modify several CUDA kernels such as the paged_attention, swap_blocks, reshape_and_cache."
> > > > > > >
> > > > > > > A minor question: what is the batch size you are using? It is currently not mentioned in the paper. I think it affects cache eviction frequency.
> > > > > > >
> > > > > > > [1] Hu, Cunchen, et al. "Memserve: Context caching for disaggregated llm serving with elastic memory pool." arXiv preprint arXiv:2406.17565 (2024).

---

> > > > > > > > ### Author Response · Authors · 2024-11-24
> > > > > > > >
> > > > > > > > Thank you again for your time and feedback. We have addressed reviewer ck7x’s comment. Please let us know if you have any questions beyond that. As for your minor question, Preble's local scheduler employs continuous batching varying based on input length, output length, and arrival time. We use the same default batch size policy as SGLang varying between 1 to max total number of tokens // 2.

---

### Meta-Review · Area_Chair_SvKM · 2024-12-21

**Metareview:**

The paper proposes "Preble," a distributed LLM serving platform optimizing prompt sharing and co-optimizing KV state reuse with computation load-balancing. The authors introduce hierarchical scheduling mechanisms and a novel algorithm, E2, which purportedly outperforms existing systems like vLLM and SG-Lang. Experiments with real-world workloads demonstrate improvements in latency and cache utilization.

Strengths: The problem addressed—efficient distributed prompt sharing in large-scale LLM serving systems—is significant and timely. The proposed scheduling algorithm is conceptually innovative, and the authors provide comprehensive experimental evaluations. The paper is well-written and supported by figures and experiments that validate its claims. Additionally, the authors’ rebuttal effectively clarified technical details and addressed reviewer concerns, demonstrating thoughtfulness and flexibility in their approach.

Weaknesses: Some concerns remain regarding novelty, as related systems like MemServe and SG-Lang address similar issues. Scalability evaluations are limited to four GPUs, which may not fully capture real-world scenarios. However, the authors provided compelling justifications for these limitations and demonstrated that the proposed method offers meaningful contributions, particularly in balancing load and optimizing prompt-sharing mechanisms.

The decision to accept is based on the paper’s potential impact on distributed LLM serving systems and its technical contributions, which outweigh the noted limitations.

**Additional Comments On Reviewer Discussion:**

The reviewers engaged in constructive discussions, and the authors responded comprehensively to concerns, including clarifying integration details and addressing scalability questions. The effort in the rebuttal phase significantly improved the overall perception of the paper, justifying its acceptance.

---

### Decision · Program_Chairs · 2025-01-22

Accept (Poster)